# Distributional Gradient Matching for Learning Uncertain Neural Dynamics Models

**Lenart Treven**[*]
ETH Zürich
trevenl@ethz.ch

**Philippe Wenk**[*]
ETH Zürich
wenkph@ethz.ch

**Florian Dörfler**
ETH Zürich
dorfler@ethz.ch

**Andreas Krause**
ETH Zürich
krausea@ethz.ch

## Abstract

Differential equations in general and neural ODEs in particular are an essential technique in continuous-time system identification. While many deterministic learning algorithms have been designed based on numerical integration via the adjoint method, many downstream tasks such as active learning, exploration in reinforcement learning, robust control, or filtering require accurate estimates of predictive uncertainties. In this work, we propose a novel approach towards estimating epistemically uncertain neural ODEs, avoiding the numerical integration bottleneck. Instead of modeling uncertainty in the ODE parameters, we directly model uncertainties in the state space. Our algorithm – *distributional gradient matching (DGM)* – jointly trains a smoother and a dynamics model and matches their gradients via minimizing a Wasserstein loss. Our experiments show that, compared to traditional approximate inference methods based on numerical integration, our approach is faster to train, faster at predicting previously unseen trajectories, and in the context of neural ODEs, significantly more accurate.

## 1 Introduction

For continuous-time system identification and control, ordinary differential equations form an essential class of models, deployed in applications ranging from robotics (Spong et al., 2006) to biology (Jones et al., 2009). Here, it is assumed that the evolution of a system is described by the evolution of continuous state variables, whose time-derivative is given by a set of parametrized equations. Often, these equations are derived from first principles, e.g., rigid body dynamics (Wittenburg, 2013), mass action kinetics (Ingalls, 2013), or Hamiltonian dynamics (Greydanus et al., 2019), or chosen for computational convenience (e.g., linear systems (Ljung, 1998)) or parametrized to facilitate system identification (Brunton et al., 2016).

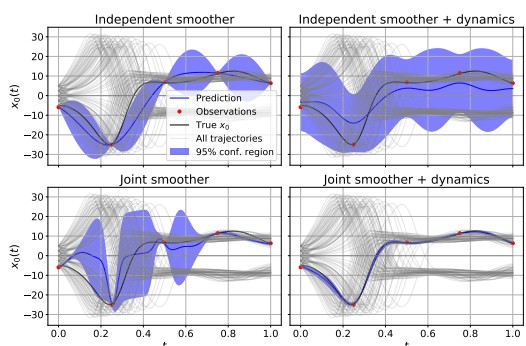

Figure 1: Illustration of DGM: Learning a joint smoother (first vs second row) across trajectories enables sharing observational data. Dynamics regularization (first vs second column) substantially improves prediction accuracy of joint smoother.

Such construction methods lead to intriguing properties, including guarantees on physical realizability (Wensing et al., 2017), favorable convergence properties (Ortega et al., 2018), or a structure suitable for downstream tasks such as control design (Ortega et al., 2002). However, such models often capture the system dynamics only approximately, leading to a potentially significant discrepancy between the model and reality (Ljung, 1999). Moreover, when expert knowledge is not available, or precise parameter values are cumbersome to obtain, system identification from raw time series data becomes

---

[*]Equal Contribution. Correspondence to trevenl@ethz.ch, wenkph@ethz.ch.

35th Conference on Neural Information Processing Systems (NeurIPS 2021).

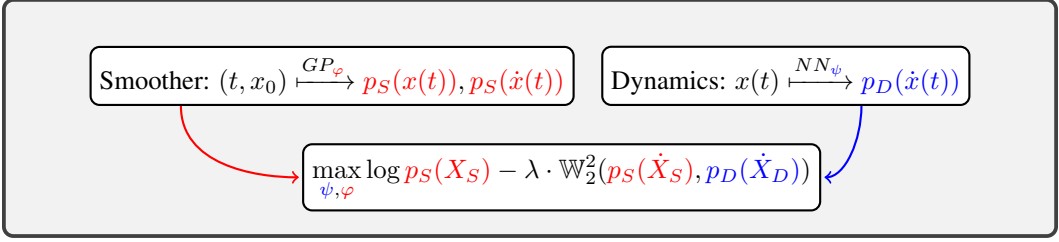

Figure 2: High-level depiction of DGM.

necessary. In this case, one may seek more expressive *nonparametric* models instead (Rackauckas et al., 2020; Pillonetto et al., 2014). If the model is completely replaced by a neural network, the resulting model is called *neural ODE* (Chen et al., 2018). Despite their large number of parameters, as demonstrated by Chen et al. (2018); Kidger et al. (2020); Zhuang et al. (2020, 2021), deterministic neural ODEs can be efficiently trained, enabling accurate deterministic trajectory predictions. For many practical applications however, accurate *uncertainty estimates* are essential, as they guide downstream tasks like reinforcement learning (Deisenroth and Rasmussen, 2011; Schulman et al., 2015), safety guarantees (Berkenkamp et al., 2017), robust control design (Hjalmarsson, 2005), planning under uncertainty (LaValle, 2006), probabilistic forecasting in meteorology (Fanfarillo et al., 2021), or active learning / experimental design (Srinivas et al., 2010). A common way of obtaining such uncertainties is via a Bayesian framework. However, as observed by Dandekar et al. (2021), Bayesian training of neural ODEs in a dynamics setting remains largely unexplored. They demonstrate that initial variational-based inference schemes for Bayesian neural ODEs suffer from several serious drawbacks and thus propose sampling-based alternatives. However, as surfaced by our experiments in Section 4, sampling-based approaches still exhibit serious challenges. These pertain both to robustness (even if highly informed priors are supplied), and reliance on frequent numerical integration of large neural networks, which poses severe computational challenges for many downstream tasks like sampling-based planning (Karaman and Frazzoli, 2011) or uncertainty propagation in prediction.

**Contributions**   In this work, we propose a novel approach for uncertainty quantification in nonlinear dynamical systems (cf. Figure 1). Crucially, our approach avoids explicit costly and non-robust numerical integration, by employing a probabilistic smoother of the observational data, whose representation we learn jointly across multiple trajectories. To capture dynamics, we regularize our smoother with a dynamics model. Latter captures epistemic uncertainty in the gradients of the ODE, which we match with the smoother's gradients by minimizing a Wasserstein loss, hence we call our approach *Distributional Gradient Matching (*DGM*)*. In summary, our main contributions are:

- We develop DGM, an approach[2] for capturing epistemic uncertainty about nonlinear dynamical systems by *jointly* training a smoother and a neural dynamics model;
- We provide a computationally efficient and statistically accurate mechanism for prediction, by focusing directly on the posterior / predictive state distribution.
- We experimentally demonstrate the effectiveness of our approach on learning challenging, chaotic dynamical systems, and generalizing to new unseen inital conditions.

**High-level overview**   A high-level depiction of our algorithm is shown in Figure 2. In principle, DGM jointly learns a smoother (S) and a dynamics model (D). The smoother model, chosen to be a Gaussian process, maps an initial condition $x_0$ and a time $t$ to the state distribution $p_S(x(t))$ and state derivatives distribution $p_S(\dot{x}(t))$ reached at that time. The dynamics model, chosen to be a neural network, represents an ODE that maps states $x(t)$ to the derivative distribution $p_D(\dot{x}(t))$. Both models are evaluated at some training times and all its output distributions collected in the random variables $X_S$, $\dot{X}_S$ and $\dot{X}_D$. The parameters of these models are then jointly trained using a Wasserstein-distance-based objective directly on the level of distributions. For more details on every one of these components, we refer to Section 3. There, we introduce all components individually and then present how they interplay. Section 3 builds on known concepts from the literature, which we

---

[2]Code is available at: `https://github.com/lenarttreven/dgm`

summarize in Section 2. Finally, in Section 4, we present the empirical study of the DGM algorithm, where we benchmark it against the state-of-the-art, uncertainty aware dynamics models.

## 2 Background

### 2.1 Problem Statement

Consider a continuous-time dynamical system whose $K$-dimensional state $\boldsymbol{x} \in \mathbb{R}^K$ evolves according to an unknown ordinary differential equation of the form

$$\dot{\boldsymbol{x}} = \boldsymbol{f}^*(\boldsymbol{x}). \tag{1}$$

Here, $\boldsymbol{f}^*$ is an arbitrary, unknown function assumed to be locally Lipschitz continuous, to guarantee existence and uniqueness of trajectories for every initial condition. In our experiment, we initialize the system at $M$ different initial conditions $\boldsymbol{x}_m(0)$, $m \in \{1, \ldots, M\}$, and let it evolve to generate $M$ trajectories. Each trajectory is then observed at discrete (but not necessarily uniformly spaced) time-points, where the number of observations $(N_m)_{m \in \{1 \ldots M\}}$ can vary from trajectory to trajectory. Thus, a trajectory $m$ is described by its initial condition $\boldsymbol{x}_m(0)$, and the observations $\boldsymbol{y}_m := [\boldsymbol{x}_m(t_{n,m}) + \boldsymbol{\epsilon}_{n,m}]_{n \in \{1 \ldots N_m\}}$ at times $\boldsymbol{t}_m := [t_{n,m}]_{n \in \{1 \ldots N_m\}}$, where the additive observation noise $\boldsymbol{\epsilon}_{n,m}$ is assumed to be drawn i.i.d. from a zero mean Gaussian, whose covariance is given by $\boldsymbol{\Sigma}_{\boldsymbol{\epsilon}} := \operatorname{diag}(\sigma_1^2, \ldots, \sigma_K^2)$. We denote by $\mathcal{D}$ the dataset, consisting of $M$ initial conditions $\boldsymbol{x}_m(0)$, observation times $\boldsymbol{t}_m$, and observations $\boldsymbol{y}_m$.

To model the unknown dynamical system, we choose a parametric Ansatz $\dot{\boldsymbol{x}} = \boldsymbol{f}(\boldsymbol{x}, \boldsymbol{\theta})$. Depending on the amount of expert knowledge, this parameterization can follow a white-box, gray-box, or black-box methodology (Bohlin, 2006). In any case, the parametric form of $\boldsymbol{f}$ is fixed a priori (e.g., a neural network), and the key challenge is to infer a reasonable distribution over the parameters $\boldsymbol{\theta}$, conditioned on the data $\mathcal{D}$. For later tasks, we are particularly interested in the *predictive posterior state distribution*

$$p(\boldsymbol{x}_{\text{new}}(\boldsymbol{t}_{\text{new}})|\mathcal{D}, \boldsymbol{t}_{\text{new}}, \boldsymbol{x}_{\text{new}}(0)), \tag{2}$$

i.e., the posterior distribution of the states starting from a potentially unseen initial condition $\boldsymbol{x}_{\text{new}}(0)$ and evaluated at times $\boldsymbol{t}_{\text{new}}$. This posterior would then be used by the downstream or prediction tasks described in the introduction.

### 2.2 Bayesian Parameter Inference

In the case of Bayesian parameter inference, an additional prior $p(\boldsymbol{\theta})$ is imposed on the parameters $\boldsymbol{\theta}$ so that the posterior distribution of Equation (2) can be inferred. Unfortunately, this distribution is not analytically tractable for most choices of $\boldsymbol{f}(\boldsymbol{x}, \boldsymbol{\theta})$, which is especially true when we model $\boldsymbol{f}$ with a neural network. Formally, for fixed parameters $\boldsymbol{\theta}$, initial condition $\boldsymbol{x}(0)$ and observation time $t$, the likelihood of an observation $\boldsymbol{y}$ is given by

$$p(\boldsymbol{y}(t)|\boldsymbol{x}(0), t, \boldsymbol{\theta}, \boldsymbol{\Sigma}_{\text{obs}}) = \mathcal{N}\left(\boldsymbol{y}(t) \middle| \boldsymbol{x}(0) + \int_0^t \boldsymbol{f}(\boldsymbol{x}(\tau), \boldsymbol{\theta}) d\tau, \boldsymbol{\Sigma}_{\text{obs}}\right). \tag{3}$$

Using the fact that all noise realizations are independent, the expression (3) can be used to calculate the likelihood of all observations in $\mathcal{D}$. Most state-of-the-art parameter inference schemes use this fact to create samples $\hat{\boldsymbol{\theta}}_s$ of the posterior over parameters $p(\boldsymbol{\theta}|\mathcal{D})$ using various Monte Carlo methods. Given a new initial condition $\boldsymbol{x}(0)$ and observation time $t$, these samples $\hat{\boldsymbol{\theta}}_s$ can then be turned into samples of the predictive posterior state again by numerically integrating

$$\hat{\boldsymbol{x}}_s(t) = \boldsymbol{x}(0) + \int_0^t \boldsymbol{f}(\boldsymbol{x}(\tau), \hat{\boldsymbol{\theta}}_s) d\tau. \tag{4}$$

Clearly, both training (i.e., obtaining the samples $\hat{\boldsymbol{\theta}}_s$) and prediction (i.e., evaluating Equation (4)) require integrating the system dynamics $\boldsymbol{f}$ many times. Especially when we model $\boldsymbol{f}$ with a neural network, this can be a huge burden, both numerically and computationally (Kelly et al., 2020).

As an alternative approach, we can approximate the posterior $p(\boldsymbol{\theta}|\mathcal{D})$ with variational inference (Bishop, 2006). However, we run into similar bottlenecks. While optimizing the variational objective, e.g., the ELBO, many integration steps are necessary to evaluate the unnormalized posterior. Also, at inference time, to obtain a distribution over state $\hat{\boldsymbol{x}}_s(t)$, we still need to integrate $\boldsymbol{f}$ several times. Furthermore, Dandekar et al. (2021) report poor forecasting performance by the variational approach.

# 3 Distributional Gradient Matching

In both the Monte Carlo sampling-based and variational approaches, all information about the dynamical system is stored in the estimates of the system parameters $\hat{\boldsymbol{\theta}}$. This makes these approaches rather cumbersome: Both for obtaining estimates of $\hat{\boldsymbol{\theta}}$ and for obtaining the predictive posterior over states, once $\hat{\boldsymbol{\theta}}$ is found, we need multiple rounds of numerically integrating a potentially complicated (neural) differential equation. We thus have identified two bottlenecks limiting the performance and applicability of these algorithms: namely, numerical integration of $\boldsymbol{f}$ and inference of the system parameters $\boldsymbol{\theta}$. In our proposed algorithm, we *avoid both* of these bottlenecks by directly working with the posterior distribution in the state space.

To this end, we introduce a probabilistic, differentiable *smoother model*, that directly maps a tuple $(t, \boldsymbol{x}(0))$ consisting of a time point $t$ and an initial condition $\boldsymbol{x}(0))$ as input and maps it to the corresponding distribution over $\boldsymbol{x}(t)$. Thus, the smoother directly replaces the costly, numerical integration steps, needed, e.g., to evaluate Equation (2).

Albeit computationally attractive, this approach has one serious drawback. Since the smoother no longer explicitly integrates differential equations, there is no guarantee that the obtained smoother model follows any vector field. Thus, the smoother model is strictly more general than the systems described by Equation (1). Unlike ODEs, it is able to capture mappings whose underlying functions violate, e.g., Lipschitz or Markovianity properties, which is clearly not desirable. To address this issue, we introduce a regularization term, $\mathcal{L}_{\text{dynamics}}$, which ensures that a trajectory predicted by the smoother is encouraged to follow some underlying system of the form of Equation (1). The smoother is then trained with the multi-objective loss function

$$\mathcal{L} \coloneqq \mathcal{L}_{\text{data}} + \lambda \cdot \mathcal{L}_{\text{dynamics}}, \tag{5}$$

where, $\mathcal{L}_{\text{data}}$ is a smoother-dependent loss function that ensures a sufficiently accurate data fit, and $\lambda$ is a trade-off parameter.

## 3.1 Regularization by Matching Distributions over Gradients

To ultimately define $\mathcal{L}_{\text{dynamics}}$, first choose a parametric *dynamics model* similar to $\boldsymbol{f}(\boldsymbol{x}, \boldsymbol{\theta})$ in Equation (3), that maps states to their derivatives. Second, define a set of *supporting points* $\mathcal{T}$ with the corresponding *supporting gradients* $\dot{\mathcal{X}}$ as

$$\mathcal{T} \coloneqq \left\{ (t_{\text{supp},l}, \boldsymbol{x}_{\text{supp},l}(0))_{l \in \{1 \ldots N_{\text{supp}}\}} \right\}, \quad \dot{\mathcal{X}} \coloneqq \left\{ (\dot{\boldsymbol{x}}_{\text{supp},l})_{l \in \{1 \ldots N_{\text{supp}}\}} \right\}.$$

Here, the $l$-th element represents the event that the dynamical system's derivative at time $t_{\text{supp},l}$ is $\dot{\boldsymbol{x}}_{\text{supp},l}$, after being initialized at time 0 at initial condition $\boldsymbol{x}_{\text{supp},l}(0)$.

Given both the smoother and the dynamics model, we have now two different ways to calculate distributions over $\dot{\mathcal{X}}$ given some data $\mathcal{D}$ and supporting points $\mathcal{T}$. First, we can directly leverage the differentiability and global nature of our smoother model to extract a distribution $p_S(\dot{\mathcal{X}}|\mathcal{D}, \mathcal{T})$ from the smoother model. Second, we can first use the smoother to obtain state estimates and then plug these state estimates into the dynamics model, to obtain a second distribution $p_D(\dot{\mathcal{X}}|\mathcal{D}, \mathcal{T})$. Clearly, if the solution proposed by the smoother follows the dynamics, these two distributions should match. Thus, we can regularize the smoother to follow the solution of Equation (3) by defining $\mathcal{L}_{\text{dynamics}}$ to encode the *distance* between $p_D(\dot{\mathcal{X}}|\mathcal{D}, \mathcal{T})$ and $p_S(\dot{\mathcal{X}}|\mathcal{D}, \mathcal{T})$ to be small in some metric. By minimizing the overall loss, we thus match the distributions over the gradients of the smoother and the dynamics model.

## 3.2 Smoothing jointly over Trajectories with Deep Gaussian Processes

The core of DGM is formed by a smoother model. In principle, the posterior state distribution of Equation (2) could be modeled by any Bayesian regression technique. However, calculating $p_S(\dot{\mathcal{X}}|\mathcal{D}, \mathcal{T})$ is generally more involved. Here, the key challenge is evaluating this posterior, which is already computationally challenging, e.g., for simple Bayesian neural networks. For Gaussian processes, however, this becomes straightforward, since derivatives of GPs remain GPs (Solak et al., 2003). Thus, DGM uses a GP smoother. For scalability and simplicity, we keep $K$ different, independent smoothers, one for each state dimension. However, if computational complexity is not a concern, our approach generalizes directly to multi-output Gaussian processes. Below, we focus on the one-dimensional case, for clarity of exposition. For notational compactness, all vectors with a

superscript should be interpreted as vectors over time in this subsection. For example, the vector $\boldsymbol{x}^{(k)}$ consists of all the $k$-th elements of the state vectors $\boldsymbol{x}(t_{n,m}), n \in \{1, \ldots, N_m\}, m \in \{1, \ldots, M\}$.

We define a Gaussian process with a differentiable mean function $\mu(\boldsymbol{x}_m(0), t_{n,m})$ as well as a differentiable and positive-definite kernel function $\mathcal{K}_{\text{RBF}}(\boldsymbol{\phi}(\boldsymbol{x}_m(0), t_{n,m}), \boldsymbol{\phi}(\boldsymbol{x}_{m'}(0), t_{n',m'})$. Here, the kernel is given by the composition of a standard ARD-RBF kernel (Rasmussen, 2004) and a differentiable feature extractor $\phi$ parametrized by a deep neural network, as introduced by Wilson et al. (2016). Following Solak et al. (2003), given fixed $\boldsymbol{x}_{\text{supp}}$, we can now calculate the joint density of $(\dot{\boldsymbol{x}}_{\text{supp}}^{(k)}, \boldsymbol{y}^{(k)})$ for each state dimension $k$. Concatenating vectors accordingly across time and trajectories, let

$$\boldsymbol{\mu}^{(k)} := \mu^{(k)}(\boldsymbol{x}(0), \boldsymbol{t}), \quad \dot{\boldsymbol{\mu}}^{(k)} := \frac{\partial}{\partial t} \mu^{(k)}(\boldsymbol{x}_{\text{supp}}(0), \boldsymbol{t}_{\text{supp}}),$$

$$\boldsymbol{z}^{(k)} := \phi^{(k)}(\boldsymbol{x}(0), \boldsymbol{t}), \quad \boldsymbol{z}_{\text{supp}}^{(k)} := \phi^{(k)}(\boldsymbol{x}_{\text{supp}}(0), \boldsymbol{t}_{\text{supp}}),$$

$$\boldsymbol{\mathcal{K}}^{(k)} := \mathcal{K}_{\text{RBF}}^{(k)}(\boldsymbol{z}^{(k)}, \boldsymbol{z}^{(k)}), \quad \dot{\boldsymbol{\mathcal{K}}}^{(k)} := \frac{\partial}{\partial t_1} \mathcal{K}_{\text{RBF}}^{(k)}(\boldsymbol{z}_{\text{supp}}^{(k)}, \boldsymbol{z}^{(k)}), \quad \ddot{\boldsymbol{\mathcal{K}}}^{(k)} := \frac{\partial^2}{\partial t_1 \partial t_2} \mathcal{K}_{\text{RBF}}^{(k)}(\boldsymbol{z}_{\text{supp}}^{(k)}, \boldsymbol{z}_{\text{supp}}^{(k)}).$$

Then the joint density of $(\dot{\boldsymbol{x}}_{\text{supp}}^{(k)}, \boldsymbol{y}^{(k)})$ can be written as

$$\begin{pmatrix} \dot{\boldsymbol{x}}_{\text{supp}}^{(k)} \\ \boldsymbol{y}^{(k)} \end{pmatrix} \sim \mathcal{N} \left( \begin{pmatrix} \dot{\boldsymbol{\mu}}^{(k)} \\ \boldsymbol{\mu}^{(k)} \end{pmatrix}, \begin{pmatrix} \ddot{\boldsymbol{\mathcal{K}}}^{(k)} & \dot{\boldsymbol{\mathcal{K}}}^{(k)} \\ (\dot{\boldsymbol{\mathcal{K}}}^{(k)})^\top & \boldsymbol{\mathcal{K}}^{(k)} + \sigma_k^2 \boldsymbol{I} \end{pmatrix} \right). \tag{6}$$

Here we denote by $\frac{\partial}{\partial t_1}$ the partial derivative with respect to time in the first coordinate, by $\frac{\partial}{\partial t_2}$ the partial derivative with respect to time in the second coordinate, and with $\sigma_k^2$ the corresponding noise variance of $\boldsymbol{\Sigma}_{\text{obs}}$.

Since the conditionals of a joint Gaussian random variable are again Gaussian distributed, $p_S$ is again Gaussian, i.e., $p_S(\dot{\mathcal{X}}_k | \mathcal{D}, \mathcal{T}) = \mathcal{N}\left(\dot{\boldsymbol{x}}_{\text{supp}}^{(k)} | \boldsymbol{\mu}_S, \boldsymbol{\Sigma}_S\right)$ with

$$\boldsymbol{\mu}_S := \dot{\boldsymbol{\mu}}^{(k)} + \dot{\boldsymbol{\mathcal{K}}}^{(k)}(\boldsymbol{\mathcal{K}}^{(k)} + \sigma_k^2 \boldsymbol{I})^{-1}\left(\boldsymbol{y}^{(k)} - \boldsymbol{\mu}^{(k)}\right),$$
$$\boldsymbol{\Sigma}_S := \ddot{\boldsymbol{\mathcal{K}}}^{(k)} - \dot{\boldsymbol{\mathcal{K}}}^{(k)}(\boldsymbol{\mathcal{K}}^{(k)} + \sigma_k^2 \boldsymbol{I})^{-1}(\dot{\boldsymbol{\mathcal{K}}}^{(k)})^\top. \tag{7}$$

Here, the index $k$ is used to highlight that this is just the distribution for one state dimension. To obtain the final $p_S(\dot{\mathcal{X}} | \mathcal{D}, \mathcal{T})$, we take the product over all state dimensions $k$.

To fit our model to the data, we minimize the negative marginal log likelihood of our observations, neglecting purely additive terms (Rasmussen, 2004), i.e.,

$$\mathcal{L}_{\text{data}} := \sum_{k=1}^{K} \frac{1}{2}\left(\boldsymbol{y}^{(k)} - \boldsymbol{\mu}^{(k)}\right)^\top \left(\boldsymbol{\mathcal{K}}^{(k)} + \sigma_k^2 \boldsymbol{I}\right)^{-1}\left(\boldsymbol{y}^{(k)} - \boldsymbol{\mu}^{(k)}\right) + \frac{1}{2} \text{logdet}\left(\boldsymbol{\mathcal{K}}^{(k)} + \sigma_k^2 \boldsymbol{I}\right). \tag{8}$$

Furthermore, the predictive posterior for a new point $x_{\text{test}}^{(k)}$ given time $t_{\text{test}}$ and initial condition $x_{\text{test}}^{(k)}(0)$ has the closed form

$$p_S(x_{\text{test}}^{(k)} | \mathcal{D}_k, t_{\text{test}}, \boldsymbol{x}_{\text{test}}) = \mathcal{N}\left(x_{\text{test}}^{(k)} \middle| \mu_{\text{post}}^{(k)}, \sigma_{\text{post},k}^2\right), \tag{9}$$

where
$$\mu_{\text{post}}^{(k)} = \mu^{(k)}(\boldsymbol{x}_{\text{test}}(0), t_{\text{test}}) + \mathcal{K}_{\text{RBF}}^{(k)}(\boldsymbol{z}_{\text{test}}^{(k)}, \boldsymbol{z}^{(k)})^\top (\boldsymbol{\mathcal{K}}^{(k)} + \sigma_k^2 \boldsymbol{I})^{-1}\left(\boldsymbol{y}^{(k)} - \boldsymbol{\mu}^{(k)}\right), \tag{10}$$

$$\sigma_{\text{post},k}^2 = \mathcal{K}_{\text{RBF}}^{(k)}(\boldsymbol{z}_{\text{test}}, \boldsymbol{z}_{\text{test}}) - \mathcal{K}_{\text{RBF}}^{(k)}(\boldsymbol{z}_{\text{test}}^{(k)}, \boldsymbol{z}^{(k)})^\top (\boldsymbol{\mathcal{K}}^{(k)} + \sigma_k^2 \boldsymbol{I})^{-1} \mathcal{K}_{\text{RBF}}^{(k)}(\boldsymbol{z}_{\text{test}}^{(k)}, \boldsymbol{z}^{(k)}). \tag{11}$$

### 3.3 Representing Uncertainty in the Dynamics Model via the Reparametrization Trick

As described at the beginning of this section, a key bottleneck of standard Bayesian approaches is the potentially high dimensionality of the dynamics parameter vector $\boldsymbol{\theta}$. The same is true for our approach. If we were to keep track of the distributions over all parameters of our dynamics model, calculating $p_D(\dot{\mathcal{X}} | \mathcal{D}, \mathcal{T})$ quickly becomes infeasible.

However, especially in the case of modeling $\boldsymbol{f}$ with a neural network, the benefits of keeping distributions directly over $\boldsymbol{\theta}$ is unclear due to overparametrization. For both the downstream tasks and our training method, we are mainly interested in the distributions in the state space. Usually, the state space is significantly lower dimensional compared to the parameter space of $\boldsymbol{\theta}$. Furthermore, since the exact posterior state distributions are generally intractable, they normally have to be approximated anyways with simpler distributions for downstream tasks (Schulman et al., 2015; Houthooft et al., 2016; Berkenkamp et al., 2017). Thus, we change the parametrization of our dynamics model as follows. Instead of working directly with $\dot{\boldsymbol{x}}(t) = \boldsymbol{f}(\boldsymbol{x}(t), \boldsymbol{\theta})$ and keeping a distribution over $\boldsymbol{\theta}$, we model uncertainty directly on the level of the vector field as

$$\dot{\boldsymbol{x}}(t) = \boldsymbol{f}(\boldsymbol{x}(t), \boldsymbol{\psi}) + \boldsymbol{\Sigma}_D^{\frac{1}{2}}(\boldsymbol{x}(t), \boldsymbol{\psi})\boldsymbol{\epsilon}, \tag{12}$$

where $\boldsymbol{\epsilon} \sim \mathcal{N}(0, \boldsymbol{I}_K)$ is drawn once per rollout (i.e., fixed within a trajectory) and $\boldsymbol{\Sigma}_D$ is a state-dependent and positive semi-definite matrix parametrized by a neural network. Here, $\boldsymbol{\psi}$ are the parameters of the new dynamics model, consisting of both the original parameters $\boldsymbol{\theta}$ and the weights of the neural network parametrizing $\boldsymbol{\Sigma}_D$. To keep the number of parameters reasonable, we employ a weight sharing scheme, detailed in Appendix B.

In spirit, this modeling paradigm is very closely related to standard Bayesian training of NODEs. In both cases, the random distributions capture a distribution over a set of deterministic, ordinary differential equations. This should be seen in stark contrast to stochastic differential equations, where the randomness in the state space, i.e., diffusion, is modeled with a stochastic process. In comparison to (12), the latter is a time-varying disturbance added to the vector field. In that sense, our model still captures the *epistemic* uncertainty about our system dynamics, while an SDE model captures the intrinsic process noise, i.e., *aleatoric* uncertainty. While this reparametrization does not allow us to directly calculate $p_D(\dot{\mathcal{X}}|\mathcal{D}, \mathcal{T})$, we obtain a Gaussian distribution for the marginals $p_D(\dot{\boldsymbol{x}}_{\text{supp}}|\boldsymbol{x}_{\text{supp}})$. To retrieve $p_D(\dot{\mathcal{X}}|\mathcal{D}, \mathcal{T})$, we use the smoother model's predictive state posterior to obtain

$$p_D(\dot{\mathcal{X}}|\mathcal{D}, \mathcal{T}) = \int p_D(\dot{\boldsymbol{x}}_{\text{supp}}, \boldsymbol{x}_{\text{supp}}|\mathcal{D}, \mathcal{T})d\boldsymbol{x}_{\text{supp}} \tag{13}$$

$$\approx \int p_D(\dot{\boldsymbol{x}}_{\text{supp}}|\boldsymbol{x}_{\text{supp}})p_S(\boldsymbol{x}_{\text{supp}}|\mathcal{T}, \mathcal{D})d\boldsymbol{x}_{\text{supp}}. \tag{14}$$

### 3.4 Comparing Gradient Distributions via the Wasserstein Distance

To compare and eventually match $p_D(\dot{\mathcal{X}}|\mathcal{D}, \mathcal{T})$ and $p_S(\dot{\mathcal{X}}|\mathcal{D}, \mathcal{T})$, we propose to use the Wasserstein distance (Kantorovich, 1939), since it allows for an analytic, closed-form representation, and since it outperforms similar measures (like forward, backward and symmetric KL divergence) in our exploratory experiments. The squared type-2 Wasserstein distance gives rise to the term

$$\mathbb{W}_2^2\left[p_S(\dot{\mathcal{X}}|\mathcal{D}, \mathcal{T}), p_D(\dot{\mathcal{X}}|\mathcal{D}, \mathcal{T})\right] = \mathbb{W}_2^2\left[p_S(\dot{\mathcal{X}}|\mathcal{D}, \mathcal{T}), \mathbb{E}_{\boldsymbol{x}_{\text{supp}} \sim p_{\text{GP}}(\boldsymbol{x}_{\text{supp}}|\mathcal{D}, \mathcal{T})}\left[p_D(\dot{\boldsymbol{x}}_{\text{supp}}|\boldsymbol{x}_{\text{supp}})\right]\right] \tag{15}$$

that we will later use to regularize the smoothing process. To render the calculation of this regularization term computationally feasible, we introduce two approximations. First, observe that an exact calculation of the expectation in Equation (15) requires mapping a multivariate Gaussian through the deterministic neural networks parametrizing $\boldsymbol{f}$ and $\boldsymbol{\Sigma}_D$ in Equation (12). To avoid complex sampling schemes, we carry out a certainty-equivalence approximation of the expectation, that is, we evaluate the dynamics model on the posterior smoother mean $\boldsymbol{\mu}_{S, \text{supp}}$. As a result of this approximation, observe that both $p_D(\dot{\mathcal{X}}|\mathcal{D}, \mathcal{T})$ and $p_S(\dot{\mathcal{X}}|\mathcal{D}, \mathcal{T})$ become Gaussians. However, the covariance structure of these matrices is very different. Since we use independent GPs for different state dimensions, the smoother only models the covariance between the state values within the same dimension, across different time points. Furthermore, since $\boldsymbol{\epsilon}$, the random variable that captures the randomness of the dynamics across all time-points, is only $K$-dimensional, the covariance of $p_D$ will be degenerate. Thus, we do not match the distributions directly, but instead match the marginals of each state coordinate at each time point independently at the different supporting time points. Hence,

using first marginalization and then the certainty equivalence, Equation (15) reduces to

$$\mathbb{W}_2^2\left[p_S(\dot{\mathcal{X}}|\mathcal{D},\mathcal{T}), p_D(\dot{\mathcal{X}}|\mathcal{D},\mathcal{T})\right] \approx \sum_{k=1}^{K}\sum_{i=1}^{|\dot{\mathcal{X}}|}\mathbb{W}_2^2\left[p_S(\dot{x}_{\mathrm{supp}}^{(k)}(t_{\mathrm{supp},i})|\mathcal{D},\mathcal{T}), p_D(\dot{x}_{\mathrm{supp}}^{(k)}(t_{\mathrm{supp},i})|\mathcal{D},\mathcal{T})\right]$$

$$\approx \sum_{k=1}^{K}\sum_{i=1}^{|\dot{\mathcal{X}}|}\mathbb{W}_2^2\left[p_S(\dot{x}_{\mathrm{supp}}^{(k)}(t_{\mathrm{supp},i})|\mathcal{D},\mathcal{T}), p_D(\dot{x}_{\mathrm{supp}}^{(k)}(t_{\mathrm{supp},i})|\boldsymbol{\mu}_{\mathrm{S,\,supp}})\right]. \qquad (16)$$

Conveniently, the Wasserstein distance can now be calculated analytically, since for two one-dimensional Gaussians $a \sim \mathcal{N}(\mu_a, \sigma_a^2)$ and $b \sim \mathcal{N}(\mu_b, \sigma_b^2)$, we have $\mathbb{W}_2^2[a,b] = (\mu_a - \mu_b)^2 + (\sigma_a - \sigma_b)^2$.

### 3.5 Final Loss Function

As explained in the previous paragraphs, distributional gradient matching trains a smoother regularized by a dynamics model. Both the parameters of the smoother $\boldsymbol{\varphi}$, consisting of the trainable parameters of the GP prior mean $\boldsymbol{\mu}$, the feature map $\phi$, and the kernel $\mathcal{K}$, and the parameters of the dynamics model $\boldsymbol{\psi}$ are trained concurrently, using the same loss function. This loss consists of two terms, of which the regularization term was already described in Equation (16). While this term ensures that the smoother follows the dynamics, we need a second term ensuring that the smoother also follows the data. To this end, we follow standard GP regression literature, where it is common to learn the GP hyperparameters by maximizing the marginal log likelihood of the observations, i.e. $\mathcal{L}_{\mathrm{data}}$ (Rasmussen, 2004). Combining these terms, we obtain the final objective

$$\mathcal{L}(\boldsymbol{\varphi}, \boldsymbol{\psi}) \coloneqq \mathcal{L}_{\mathrm{data}} - \lambda \cdot \sum_{k=1}^{K}\sum_{i=1}^{|\dot{\mathcal{X}}|}\mathbb{W}_2^2\left[p_S(\dot{x}_{\mathrm{supp}}^{(k)}(t_{\mathrm{supp},i})|\mathcal{D},\mathcal{T}), p_D(\dot{x}_{\mathrm{supp}}^{(k)}(t_{\mathrm{supp},i})|\boldsymbol{\mu}_{\mathrm{S,\,supp}})\right].$$

This loss function is a multi-criteria objective, where fitting the data (via the smoother) and identifying the dynamics model (by matching the marginals) regularize each other. In our preliminary experiments, we found the objective to be quite robust w.r.t. different choices of $\lambda$. In the interest of simplicity, we thus set it in all our experiments in Section 4 to a default value of $\lambda = \frac{|\mathcal{D}|}{|\dot{\mathcal{X}}|}$, accounting only for the possibility of having different numbers of supporting points and observations. One special case worth mentioning is $\lambda \to 0$, which corresponds to conventional sequential smoothing, where the second part would be used for identification in a second step, as proposed by Pillonetto and De Nicolao (2010). However, as can be seen in Figure 1, the smoother fails to properly identify the system without any knowledge about the dynamics and thus fails to provide meaningful state or derivative estimates. Thus, especially in the case of sparse observations, joint training is strictly superior.

In its final form, unlike its pure Bayesian counterparts, DGM does not require any prior knowledge about the system dynamics. Nevertheless, if some prior knowledge is available, one could add an additional, additive term $\log(p(\boldsymbol{\psi}))$ to $\mathcal{L}(\boldsymbol{\varphi}, \boldsymbol{\psi})$. It should be noted however that this was not done in any of our experiments, and excellent performance can be achieved without.

## 4 Experiments

We now compare DGM against state-of-the-art methods. In a first experiment, we demonstrate the effects of an overparametrized, simple dynamics model on the performance of DGM as well as traditional, MC-based algorithms SGLD (Stochastic Gradient Lengevin Dynamics, (Welling and Teh, 2011)) and SGHMC (Stochastic Gradient Hamiltonian Monte Carlo, (Chen et al., 2014)). We select our baselines based on the results of Dandekar et al. (2021), who demonstrate that both a variational approach and NUTS (No U-Turn Sampler, Hoffman and Gelman (2014)) are inferior to these two. Subsequently, we will investigate and benchmark the ability of DGM to correctly identify neural dynamics models and to generalize across different initial conditions. Since SGLD and SGHMC reach their computational limits in the generalization experiments, we compare against Neural ODE Processes (NDP). Lastly, we will conclude by demonstrating the necessity of all of its components. For all comparisons, we use the julia implementations of SGLD and SGHMC provided by Dandekar et al. (2021), the pytorch implementation of NDP provided by Norcliffe et al. (2021), and our own JAX (Bradbury et al., 2018) implementation of DGM.

## 4.1 Setup

We use known parametric systems from the literature to generate simulated, noisy trajectories. For these benchmarks, we use the two-dimensional *Lotka Volterra (LV)* system, the three-dimensional, chaotic *Lorenz (LO)* system, a four-dimensional *double pendulum (DP)* and a twelve-dimensional *quadrocopter (QU)* model. For all systems, the exact equations and ground truth parameters are provided in the Appendix A. For each system, we create two different data sets. In the first, we include just one densely observed trajectory, taking the computational limitations of the benchmarks into consideration. In the second, we include many, but sparsely observed trajectories (5 for LV and DP, 10 for LO, 15 for QU). This setting aims to study generalization over different initial conditions.

## 4.2 Metric

We use the log likelihood as a metric to compare the accuracy of our probabilistic models. In the 1-trajectory setting, we take a grid of 100 time points equidistantly on the training trajectory. We then calculate the ground truth and evaluate its likelihood using the predictive distributions of our models. When testing for generalization, we repeat the same procedure for unseen initial conditions.

## 4.3 Effects of Overparametrization

We first study a three-dimensional, linear system of the form $\dot{\boldsymbol{x}}(t) = \boldsymbol{A}\boldsymbol{x}(t)$, where $\boldsymbol{A}$ is a randomly chosen matrix with one stable and two marginally stable eigenvalues. For the dynamics model, we choose a linear Ansatz $\boldsymbol{f}(\boldsymbol{x}, \boldsymbol{\theta}) = \boldsymbol{B}\boldsymbol{x}(t)$, where $\boldsymbol{B}$ is parametrized as the product of multiple matrices. The dimension of the matrices of each factorization are captured in a string of integers of the form $(3, a_1, \ldots, a_J, 3)$. For example, $(3, 3)$ corresponds to $\boldsymbol{B}$ being just one matrix, while $(3, 6, 6, 3)$ corresponds to $\boldsymbol{B} = \boldsymbol{B}_1 \boldsymbol{B}_2 \boldsymbol{B}_3$, with $\boldsymbol{B}_1 \in \mathbb{R}^{3 \times 6}$, $\boldsymbol{B}_2 \in \mathbb{R}^{6 \times 6}$ and $\boldsymbol{B}_3 \in \mathbb{R}^{6 \times 3}$. All of these models can be interpreted as linear neural networks, forming a simple case of the nonparametric systems we study later. Unlike general neural networks, the expressiveness of the Ansatz is independent of the number of parameters, allowing us to isolate the effects of overparametrization. In Figure 3, we show the mean and standard deviation of the log likelihood of the ground truth over 10 different noise realizations. The exact procedure for one noise realization is described in the appendix, Appendix C). While SGLD runs into numerical issues after a medium model complexity, the performance of SGHMC continuously disintegrates, while DGM is unaffected. This foreshadows the results of the next two experiments, where we observe that the MC-based approaches are not suitable for the more complicated settings.

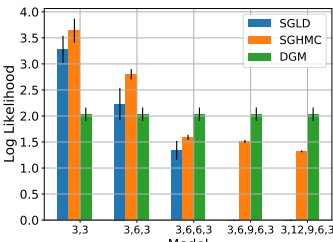

Figure 3: SGLD does not converge for strongly overparametrized models, the performance of SGHMC deteriorates. DGM is not noticeably affected.

Table 1: Log likelihood and prediction times of 100 ground truth sample points, with mean and standard deviation taken over 10 independent noise realizations, for neural ODEs trained on a single, densely sampled trajectory.

|  | Log Likelihood | | | Prediction time [ms] | | |
|---|---|---|---|---|---|---|
|  | DGM | SGHMC | SGLD | DGM | SGHMC | SGLD |
| LV 1 | $\mathbf{1.96 \pm 0.21}$ | $1.36 \pm 0.0693$ | $1.03 \pm 0.0581$ | $\mathbf{0.68 \pm 0.04}$ | $14.98 \pm 0.23$ | $14.59 \pm 0.15$ |
| LO 1 | $\mathbf{-0.57 \pm 0.11}$ | $-3.02 \pm 0.158$ | $-2.67 \pm 0.367$ | $\mathbf{0.99 \pm 0.05}$ | $98.93. \pm 5.79$ | $105.03 \pm 12.22$ |
| DP 1 | $\mathbf{2.13 \pm 0.14}$ | $1.88 \pm 0.0506$ | $1.85 \pm 0.0501$ | $\mathbf{1.31 \pm 0.05}$ | $10.60 \pm 0.21$ | $11.34 \pm 0.76$ |
| QU 1 | $\mathbf{0.64 \pm 0.07}$ | $-5.00 \pm 1.36$ | NaN | $\mathbf{3.76 \pm 0.12}$ | $24.68 \pm 6.58$ | NaN |

## 4.4 Single Trajectory Benchmarks

In Table 1, we evaluate the log-likelihood of the ground truth for the four benchmark systems, obtained when learning these systems using a neural ODE as a dynamics model (for more details, see appendix B). Clearly, DGM performs the best on all systems, even though we supplied both SGLD and SGHMC with very strong priors and fine-tuned them with an extensive hyperparameter sweep (see Appendix C for more details). Despite this effort, we failed to get SGLD to work on Quadrocopter 1, where it always returned NaNs. This is in stark contrast to DGM, which performs reliably without any pre-training or priors.

## 4.5 Prediction speed

To evaluate prediction speed, we consider the task of predicting 100 points on a previously unseen trajectory. To obtain a fair comparison, all algorithms' prediction routines were implemented in JAX (Bradbury et al., 2018). Furthermore, while we used 1000 MC samples when evaluating the predictive posterior for the log likelihood to guarantee maximal accuracy, we only used 200 samples in Table 1. Here, 200 was chosen as a minimal sample size guaranteeing reasonable accuracy, following a preliminary experiment visualized in Appendix C. Nevertheless, the predictions of DGM are 1-2 orders of magnitudes faster, as can be seen in Table 1. This further illustrates the advantage of relying on a smoother instead of costly, numerical integration to obtain predictive posteriors in the state space.

Table 2: Log likelihood of 100 ground truth sample points, with mean and covariance taken over 10 independent noise realizations, for neural ODEs trained on a multiple, sparsely sampled trajectory.The number following the system name denotes the number of trajectories in the training set.

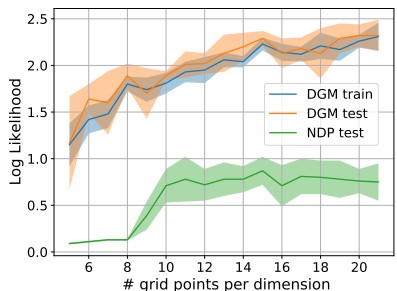

Figure 4: Log likelihood of the ground truth for Lotka Volterra for increasing number of trajectories with 5 observations each.

|  | Log Likelihood | |
|---|---|---|
|  | DGM | NDP |
| LV 100 | $\mathbf{1.81 \pm 0.08}$ | $0.62 \pm 0.27$ |
| LO 125 | $\mathbf{-2.18 \pm 0.76}$ | $-2.85 \pm 0.05$ |
| DP 100 | $\mathbf{1.86 \pm 0.05}$ | $0.88 \pm 0.05$ |
| QU 64 | $\mathbf{-0.54 \pm 0.36}$ | $-0.91 \pm 0.07$ |

## 4.6 Multi-Trajectory Benchmarks

Next, we take a set of trajectories starting on an equidistant grid of the initial conditions. Each trajectory is then observed at 5 equidistant observation times for LV and DP, and 10 equidistant observation times for the chaotic Lorenz and more complicated Quadrocopter. We test generalization by randomly sampling a new initial condition and evaluating the negative log likelihood of the ground truth at 100 equidistant time points. In Table 2, we compare the generalization performance of DGM against NDP, since despite serious tuning efforts, the MC methods failed to produce meaningful results in this setting. DGM clearly outperforms NDP, a fact which is further exemplified in Figure 4. There, we show the test log likeliood for Lotka Volterra trained on an increasing set of trajectories. Even though the time grid is fixed and we only decrease the distance between initial condition samples, the dynamics model helps the smoother to generalize across time as well. In stark contrast, NDP fails to improve with increasing data after an initial jump.

## 4.7 Ablation study

We next study the importance of different elements of our approach via an ablation study on the Lorenz 125 dataset, shown in Figure 1. Comparing the two rows, we see that joint smoothing across trajectories is essential to transfer knowledge between different training trajectories. Similarly, comparing the two columns, we see that the dynamics model enables the smoother to reduce its uncertainty in between observation points.

## 4.8 Computational Requirements

For the one trajectory setting, all DGM related experiments were run on a Nvidia RTX 2080 Ti, where the longest ones took 15 minutes. The comparison methods were given 24h, on Intel Xeon Gold 6140 CPUs. For the multi-trajectory setting, we used Nvidia Titan RTX, where all experiments finished in less than 3 hours. A more detailed run time compilation can be found in Appendix B. Using careful implementation, the run time of DGM scales linearly in the number of dimensions $K$. However, since we use an accurate RBF kernel for all our experiments reported in this section, we have cubic run time complexity in $\sum_{m=1}^{M} N_m$. In principle, this can be alleviated by deploying standard feature approximation methods (Rahimi et al., 2007; Liu et al., 2020). While this is a well known fact, we nevertheless refer the interested reader to a more detailed discussion of the subject in Appendix D.

# 5 Related work

## 5.1 Bayesian Parameter Inference with Gaussian Processes

The idea of matching gradients of a (spline-based) smoother and a dynamics model goes back to the work of Varah (1982). For GPs, this idea is introduced by Calderhead et al. (2009), who first fit a GP to the data and then match the parameters of the dynamics. Dondelinger et al. (2013) introduce concurrent training, while Gorbach et al. (2017) introduce an efficient variational inference procedure for systems with a locally-linear parametric form. All these works claim to match the distributions of the gradients of the smoother and dynamics models, by relying on a product of experts heuristics. However, Wenk et al. (2019) demonstrate that this product of experts in fact leads to statistical independence between the observations and the dynamics parameters, and that these algorithms essentially match *point estimates* of the gradients instead. Thus, DGM is the first algorithm to actually match gradients on the level of distributions for ODEs. In the context of stochastic differential equations (SDEs) with constant diffusion terms, Abbati et al. (2019) deploy MMD and GANs to match their gradient distributions. However, it should be noted that their algorithm treats the parameters of the dynamics model *deterministically* and thus, they can not provide the epistemic uncertainty estimates that we seek here. Note that our work is *not* related to the growing literature investigating SDE approximations of Bayesian Neural ODEs in the context of classification (Xu et al., 2021). Similarly to Chen et al. (2018), these works emphasize learning a terminal state of the ODE used for other downstream tasks.

## 5.2 Gaussian Processes with Operator Constraints

Gradient matching approaches mainly use the smoother as a proxy to infer dynamics parameters. This is in stark contrast to our work, where we treat the smoother as the main model used for prediction. While the regularizing properties of the dynamics on the smoother are explored by Wenk et al. (2020), Jidling et al. (2017) introduce an algorithm to incorporate linear operator constraints directly on the kernel level. Unlike in our work, they can provide strong guarantees that the posterior always follows these constraints. However, it remains unclear how to generalize their approach to the case of complex, nonlinear operators, potentially parametrized by neural dynamics models.

## 5.3 Other Related Approaches

In some sense, the smoother is mimicking a probabilistic numerical integration step, but without explicitly integrating. In spirit, this approach is similar to the solution networks used in the context of PDEs, as presented by Raissi et al. (2019), which however typically disregard uncertainty. In the context of classical ODE parameter inference, Kersting et al. (2020) deploy a GP to directly mimic a numerical integrator in a probabilistic, differentiable manner. Albeit promising in a classical, parametric ODE setting, it remains unclear how these methods can be scaled up, as there is still the numerical integration bottleneck. Unrelated to their work, Ghosh et al. (2021) present a variational inference scheme in the same, classical ODE setting. However, they still keep distributions over all weights of the neural network (Norcliffe et al., 2021). A similar approach is investigated by Dandekar et al. (2021), who found it to be inferior to the MC methods we use as a benchmark. Variational inference was previously employed by Yildiz et al. (2019) in the context of latent neural ODEs parametrized by a Bayesian neural network, but their work mainly focuses on dimensionality reduction. Nevertheless, their work inspired a model called Neural ODE Processes by Norcliffe et al. (2021). This work is similar to ours in the sense that it avoids keeping distributions over network weights and models an ensemble of deterministic ODEs via a global context variable. Consequently, we use it as a benchmark in Section 4, showing that it does not properly capture epistemic uncertainty in a low data setting, which might be problematic for downstream tasks like reinforcement learning.

# 6 Conclusion

In this work, we introduced a novel, GP-based collocation method, that matches gradients of a smoother and a dynamics model on the distribution level using a Wasserstein loss. Through careful parametrization of the dynamics model, we manage to train complicated, neural ODE models, where state of the art methods struggle. We then demonstrate that these models are able to accurately predict unseen trajectories, while capturing epistemic uncertainty relevant for downstream tasks. In future work, we are excited to see how our training regime can be leveraged in the context of active learning of Bayesian neural ordinary differential equation for continuous-time reinforcement learning.

## Acknowledgments

This research was supported by the Max Planck ETH Center for Learning Systems. This project has received funding from the European Research Council (ERC) under the European Union's Horizon 2020 research and innovation programme grant agreement No 815943 as well as from the Swiss National Science Foundation under NCCR Automation, grant agreement 51NF40 180545.

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
