# Contents of Appendix

# A Dataset description

In this section, we describe the datasets we use in our experiments.

## A.1 Lotka Volterra

The two dimensional Lotka Volterra system is governed by the parametric differential equations

$$\frac{dx}{dt} = \alpha x - \beta xy$$

$$\frac{dy}{dt} = \delta xy - \gamma y,$$

where we selected $(\alpha, \beta, \gamma, \delta) = (1, 1, 1, 1)$. These equations were numerically integrated to obtain a ground truth, where the initial conditions and observation times depend on the dataset. All observations were then created by adding additive, i.i.d. noise, distributed according to a normal distribution $\mathcal{N}\left(0, 0.1^2\right)$.

LV 1 consists of one trajectory starting from initial condition $(1, 2)$. The trajectory is observed at 100 equidistant time points from the interval $(0, 10)$.

LV 100 consists of 100 trajectories. Initial conditions for these trajectories are located on a grid, i.e.,

$$\left\{ \left( \frac{1}{2} + \frac{i}{9}, \frac{1}{2} + \frac{j}{9} \right) \middle| i \in \{0, \ldots, 9\}, j \in \{0, \ldots, 9\} \right\}.$$

Each trajectory is then observed at 5 equidistant time points from the interval $(0, 10)$, which leads to a total of 500 observations.

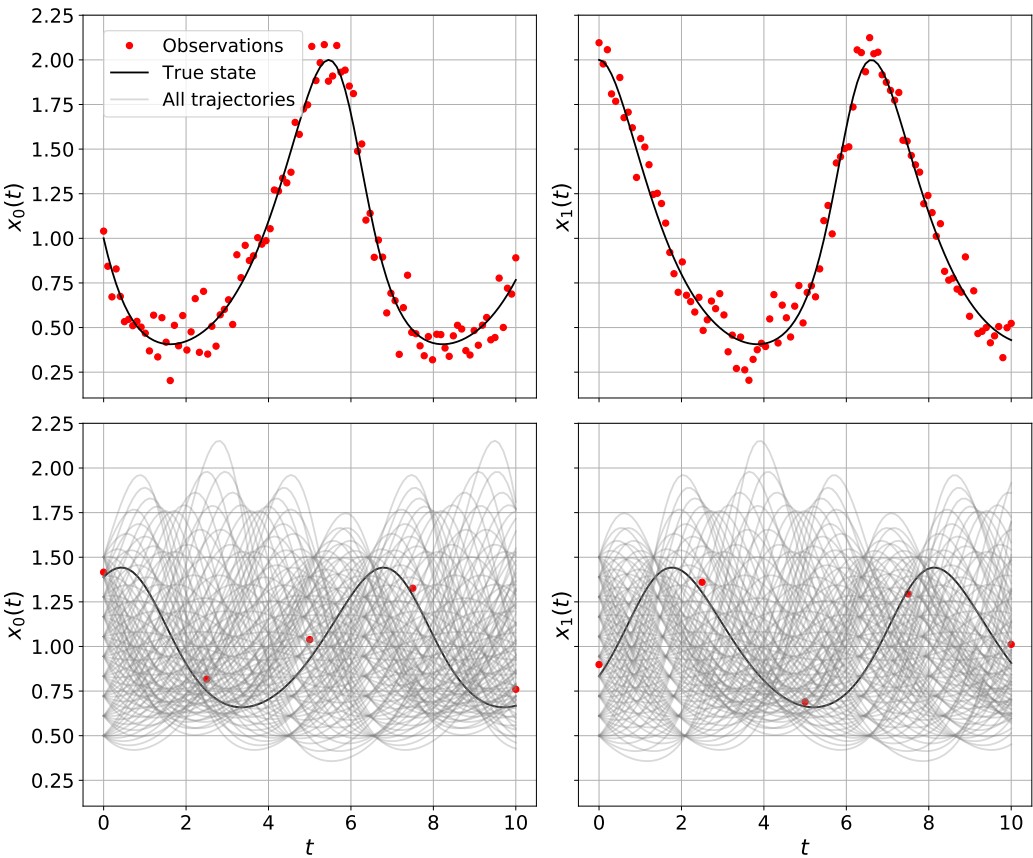

Figure 5: The first row represents the true states and all observations of LV 1 with random seed 0. In the second row we plot all ground truth trajectories from the dataset LV 100. One particular trajectory is highlighted in black, together with the corresponding observations of that trajectory (red dots).

To test generalization, we created 10 new trajectories. The initial conditions of these trajectories were obtained by sampling uniformly at random on $[0.5, 1.5]^2$. To evaluate the log likelihood, we used 100 equidistant time points from the interval $(0, 10)$.

## A.2 Lorenz

The 3 dimensional, chaotic Lorenz system is governed by the parametric differential equations

$$\frac{dx}{dt} = \sigma(y - x)$$
$$\frac{dy}{dt} = x(\rho - z) - y$$
$$\frac{dz}{dt} = xy - \tau y,$$

where we selected $(\sigma, \rho, \tau) = (10, 28, 8/3)$. These equations were numerically integrated to obtain a ground truth, where the initial conditions and observation times depend on the dataset. All observations were then created by adding additive, i.i.d. noise, distributed according to a normal distribution $\mathcal{N}(0, 1)$.

LO 1 consists of one trajectory starting from initial condition $(-2.5, 2.5, 2.5)$. The trajectory is observed at 100 equidistant time points from the interval $(0, 1)$.

LO 125 consists of 125 trajectories. Initial conditions for these trajectories are located on a grid, i.e.,

$$\left\{ \left( -5 + \frac{5i}{2}, -5 + \frac{5j}{2}, -5 + \frac{5k}{2} \right) \Big| i \in \{0, \ldots, 4\}, j \in \{0, \ldots, 4\}, k \in \{0, \ldots, 4\} \right\}.$$

Each trajectory is then observed on 10 equidistant time points from the interval $(0, 1)$, which leads to a total of 1250 observations.

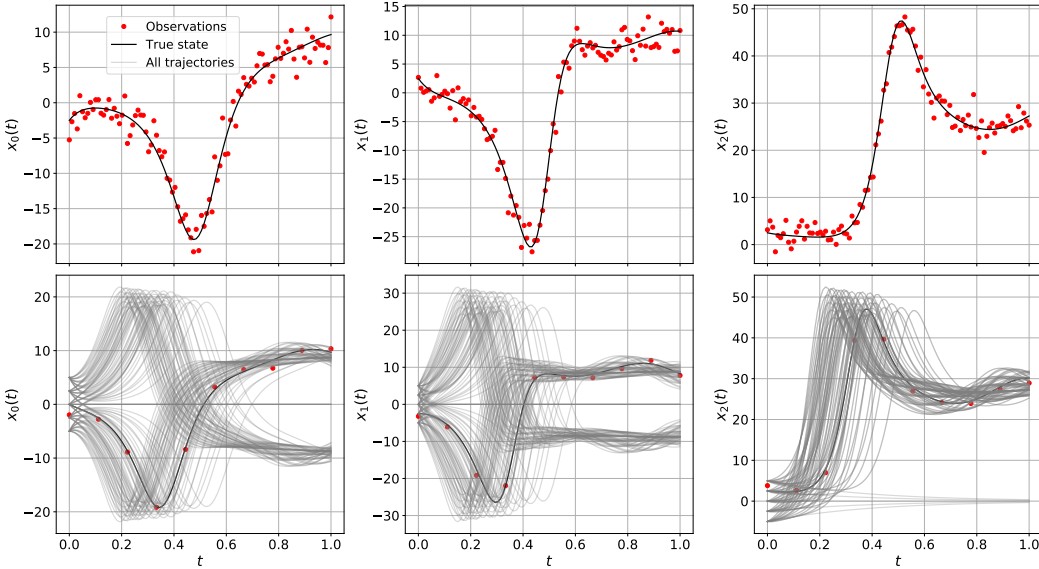

Figure 6: The first row represents the true states and all observations of LO 1 with random seed 0. In the second row we plot all ground truth trajectories from the dataset LO 125. One particular trajectory is highlighted in black, together with the corresponding observations of that trajectory (red dots).

To test generalization, we created 10 new trajectories. The initial conditions of these trajectories were obtained by sampling uniformly at random on $[-5, 5]^3$. To evaluate the log likelihood, we used 100 equidistant time points from the interval $(0, 1)$.

### A.3 Double Pendulum

The 4 dimensional Double pendulum system is governed by the parametric differential equations

$$\dot{\theta}_1 = \frac{6}{ml^2} \frac{2p_{\theta_1} - 3\cos(\theta_1 - \theta_2)p_{\theta_2}}{16 - 9\cos^2(\theta_1 - \theta_2)}$$

$$\dot{\theta}_2 = \frac{6}{ml^2} \frac{8p_{\theta_2} - 3\cos(\theta_1 - \theta_2)p_{\theta_1}}{16 - 9\cos^2(\theta_1 - \theta_2)}.$$

$$\dot{p}_{\theta_1} = -\frac{1}{2}ml^2 \left( \dot{\theta}_1\dot{\theta}_2 \sin(\theta_1 - \theta_2) + 3\frac{g}{l}\sin\theta_1 \right)$$

$$\dot{p}_{\theta_2} = -\frac{1}{2}ml^2 \left( -\dot{\theta}_1\dot{\theta}_2 \sin(\theta_1 - \theta_2) + \frac{g}{l}\sin\theta_2 \right),$$

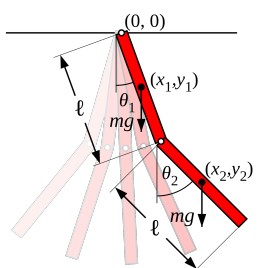

Figure 7: Double Pendulum where both rods have equal length and mass.

where we selected $(g, m, l) = (9.81, 1, 1)$. In these equations, $\theta_1$ and $\theta_2$ represent the offset angles, while $p_{\theta_1}$ and $p_{\theta_1}$ represent the momentum of the upper and lower pendulum. These equations were numerically integrated to obtain a ground truth, where the initial conditions and observation times depend on the dataset. All observations were then created by adding additive, i.i.d. noise, distributed according to a normal distribution $\mathcal{N}\left(0, 0.1^2\right)$.

DP 1 consist of one trajectory starting from initial condition $(-\pi/6, -\pi/6, 0, 0)$. The trajectory is observed at 100 equidistant time points from the interval $(0, 1)$.

DP 100 consists of 100 trajectories. Initial conditions for these trajectories are located on a grid, i.e.,

$$\left\{ \left( -\frac{\pi}{6} + \frac{\pi i}{27}, -\frac{\pi}{6} + \frac{\pi j}{27}, 0, 0 \right) \Big| i \in \{0, \ldots, 9\}, j \in \{0, \ldots, 9\} \right\}.$$

Each trajectory is then observed at 5 equidistant time points from the interval $(0, 1)$, which leads to a total of 500 observations.

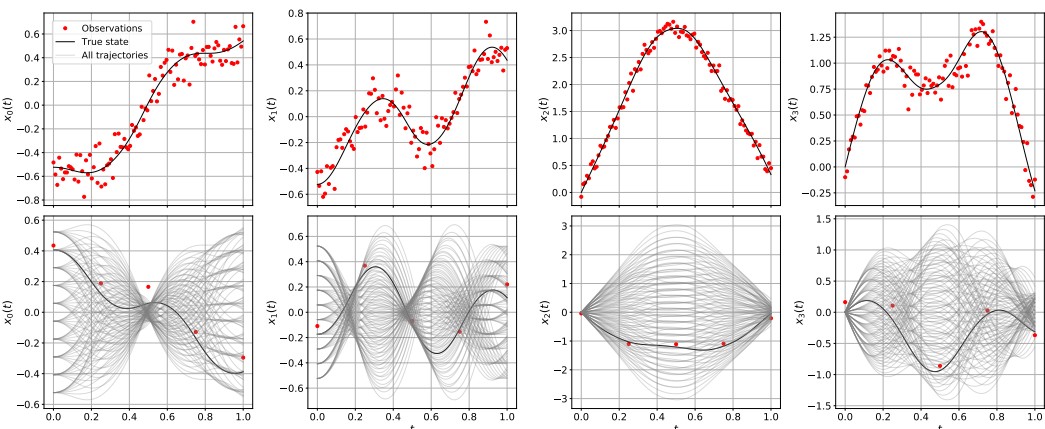

Figure 8: The first row represents the true states and all observations of DP 1 with random seed 0. In the second row we plot all ground truth trajectories from the dataset DP 100. One particular trajectory is highlighted in black, together with the corresponding observations of that trajectory (red dots).

To test generalization, we created 10 new trajectories. The initial conditions of these trajectories were obtained by sampling uniformly at random on $[-\frac{\pi}{6}, \frac{\pi}{6}]^2 \times \{0\}^2$. To evaluate the log likelihood, we used 100 equidistant time points from the interval $(0, 1)$.

### A.4 Quadrocopter

The 12 dimensional Quadrocopter system is governed by the parametric differential equations

$$\dot{u} = -g\sin(\theta) + rv - qw$$
$$\dot{v} = g\sin(\phi)\cos(\theta) - ru + pw$$
$$\dot{w} = -F_z/m + g\cos(\phi)\cos(\theta) + qu - pv$$
$$\dot{p} = \left(L + (I_{yy} - I_{zz})qr\right)/I_{xx}$$
$$\dot{q} = \left(M + (I_{zz} - I_{xx})pr\right)/I_{yy}$$
$$\dot{r} = (I_{xx} - I_{yy})pq/I_{zz}$$
$$\dot{\phi} = p + (q\sin(\phi) + r\cos(\phi))\tan(\theta)$$
$$\dot{\theta} = q\cos(\phi) - r\sin(\phi)$$
$$\dot{\psi} = (q\sin(\phi) + r\cos(\phi))\sec(\theta)$$
$$\dot{x} = \cos(\theta)\cos(\psi)u + (-\cos(\phi)\sin(\psi) + \sin(\phi)\sin(\theta)\cos(\psi))v +$$
$$+ (\sin(\phi)\sin(\psi) + \cos(\phi)\sin(\theta)\cos(\phi))w$$
$$\dot{y} = \cos(\theta)\sin(\psi)u + (\cos(\phi)\cos(\psi) + \sin(\phi)\sin(\theta)\sin(\psi))v +$$
$$+ (-\sin(\phi)\cos(\psi) + \cos(\phi)\sin(\theta)\sin(\phi))w$$
$$\dot{z} = \sin(\theta)u - \sin(\phi)\cos(\theta)v - \cos(\phi)\cos(\theta)w,$$

where

$$F_z = F_1 + F_2 + F_3 + F_4$$
$$L = (F_2 + F_3)d_y - (F_1 + F_4)d_x$$
$$M = (F_1 + F_3)d_x - (F_2 + F_4)d_x.$$

We fixed the input control forces to $(F_1, F_2, F_3, F_4) = (0.496, 0.495, 0.4955, 0.4955)$ (not to be inferred) and selected $(m, I_{xx}, I_{yy}, I_{zz}, d_x, d_y, g) = (0.1, 0.62, 1.13, 0.9, 0.114, 0.0825, 9.85)$. These equations were numerically integrated to obtain a ground truth, where the initial conditions and observation times depend on the dataset. All observations were then created by adding additive, i.i.d. noise, distributed according to a normal distribution $\mathcal{N}(0, \Sigma)$, where

$$\Sigma = \mathrm{diag}(1, 1, 1, 0.1, 0.1, 0.1, 1, 0.1, 0.1, 5, 5, 5).$$

QU 1 consists of one trajectory starting from initial condition $(0, 0, 0, 0, 0, 0, 0, 0, 0, 0, 0, 0)$. The trajectory is observed at $100$ equidistant time points from the interval $(0, 10)$.

QU 64 consists of 64 trajectories. Initial conditions for these trajectories are located on a grid, i.e.,

$$\left\{ \left(0, 0, 0, 0, 0, 0, -\frac{\pi}{18} + \frac{\pi i}{27}, -\frac{\pi}{18} + \frac{\pi j}{27}, -\frac{\pi}{18} + \frac{\pi k}{27}, 0, 0, 0\right) \,\middle|\, (i, j, k) \in \{0, \ldots, 4\}^3 \right\}.$$

Each trajectory is then observed at $15$ equidistant time points from the interval $(0, 10)$, which leads to a total of $960$ observations.

To test generalization, we created 10 new trajectories. The initial conditions of these trajectories were obtained by sampling uniformly at random on $\{0\}^6 \times [-\frac{\pi}{18}, \frac{\pi}{18}]^2 \times \{0\}^3$. To evaluate the log likelihood, we used $100$ equidistant time points from the interval $(0, 10)$.

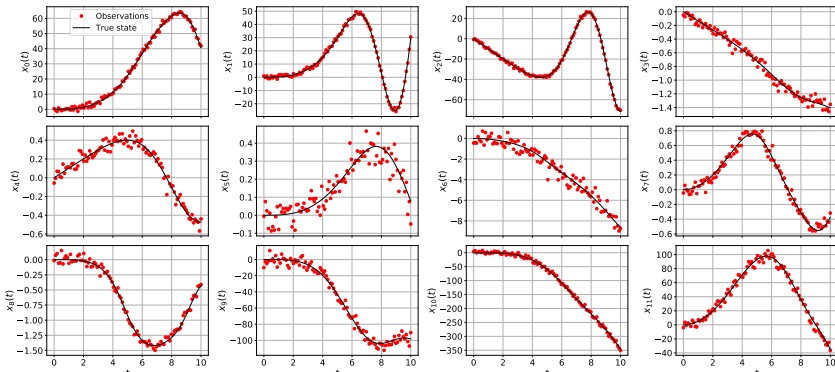

Figure 9: Visualization showing the true states and all observations of QU 1 with random seed 0.

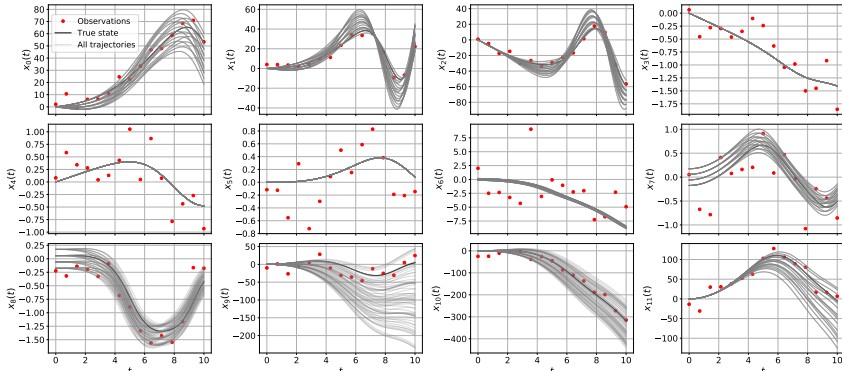

Figure 10: Visualization showing all ground truth trajectories from the dataset QU 64. One particular trajectory is highlighted in black, together with the corresponding observations of that trajectory (red dots).

# B  Implementation details of DGM

In this section we discuss all implementation details of DGM. As described in Section 3, DGM consists of a smoother and a dynamics model. At training time, the dynamics model is then used to regularize the smoother via the squared type-2 Wasserstein distance.

## B.1  Dynamics model

The role of the dynamics model is to map every state $\boldsymbol{x}$ to a distribution over derivatives, which we decide to parameterize as $\mathcal{N}\left(\boldsymbol{f}(\boldsymbol{x}, \boldsymbol{\psi}), \boldsymbol{\Sigma}_D(\boldsymbol{x}, \boldsymbol{\psi})\right)$. In the paper we focused in the case where we do not have any prior knowledge about the dynamics and we model both $\boldsymbol{f}$ and $\boldsymbol{\Sigma}_D$ with neural network. Nevertheless, if some prior knowledge about the dynamics $\boldsymbol{f}$ is available, it can be used to inform the mean function $\boldsymbol{f}$ (potentially also covariance $\boldsymbol{\Sigma}_D$) appropriately. In particular, if the parametric form of the dynamics is known, we can use them as a direct substitute for $\boldsymbol{f}$. This is the special case of ODE-parameter inference, which we investigate empirically in Appendix E. Next we present implementation details of both non-parametric (i.e. $\boldsymbol{f}$ given by a neural network) and parametric (i.e. $\boldsymbol{f}$ given by some parametric form) dynamics models.

**Non-parametric dynamics**   In the non-parametric case, we model both the dynamics' mean function $\boldsymbol{\mu}_D$ and covariance function $\boldsymbol{\Sigma}_D$ with a neural networks. Across all experiments, we choose a simple 3-layer neural network with 20, 20 and $2n$ nodes, where $n$ denotes the number of state dimensions of a specific system. After each layer, we apply the sigmoid activation function, except for the last one. The first $n$ output nodes represent the mean function. Here, no activation function is necessary for the last layer. The second $n$ output nodes are used to construct the diagonal covariance

$\boldsymbol{\Sigma}_D$. To ensure positivity, we use $x \mapsto \log(1 + \exp(x))^2$ as an activation function on the last $n$ nodes of the last layer.

**Parametric dynamics**  In the parametric case, we model $\boldsymbol{\mu}_D$ using the parametric form of the vector field. Across all experiments, we choose a simple 3-layer neural network with 10, 10 and $n$ nodes, where $n$ denotes the number of state dimensions of a specific system. After each lyer, we apply the sigmoid activation function, except for the last one. The $n$ nodes are then used to construct the diagonal covariance $\boldsymbol{\Sigma}_D$. To ensure positivity, we use $x \mapsto \log(1 + \exp(x))^2$ as an activation function on the last layer.

## B.2   Smoother model

The role of the smoother model is to map every tuple $(\boldsymbol{x}(0), t)$ consisting of initial condition $\boldsymbol{x}(0)$ and time $t$ to $\boldsymbol{x}(t)$, which is the state at time $t$ of a trajectory starting at $\boldsymbol{x}(0)$ at time 0. In the paper, we model the smoother using a Gaussian process with a deep mean function $\boldsymbol{\mu}$ and a deep feature map $\phi$. Both of them take as input the tuple $(\boldsymbol{x}(0), t)$. This tuple is then mapped through a dense neural network we call core. For all experiments, we chose a core with two layers, with 10 and 5 hidden nodes and sigmoid activation on both. The output of the core is then fed into two linear heads. The head for $\boldsymbol{\mu}$ builds a linear combination of the core's output to obtain a vector of the same shape as $\boldsymbol{x}(t)$. The head for $\phi$ builds a linear combination of the core's output to obtain a vector of length 3, the so called features. These features are then used as inputs to a standard RBF kernel with ARD (Rasmussen, 2004). For each state dimension, we keep a separate $\phi$-head, as well as separate kernel hyperparameters. However, the core is shared across dimensions, while $\boldsymbol{\mu}$ is directly introduced as multidimensional.

In the paper, we set the variance of the RBF to 1 and learned the lengthscales together with all other hyperparameters. However, due to the expressiveness of the neural network, the lengthscales are redundant and could easily be incorporated into the linear combination performed by the head. Thus, in the scaling experiments, we fix the lengthscale to one and approximate the RBF kernel with a feature expansion, as detailed in Appendix D.

## B.3   Evaluation metric

To evaluate the quality of our models' predictions, we use the log likelihood. To obtain the log likelihood, we first use the model to predict the mean and standard deviation at 100 equidistant times. Then we calculate the log likelihood of the ground truth for every predicted point. We take the mean over dimensions, over times, and over trajectories. When reporting the training log likelihood, as done e.g. in Table 1, we use the training trajectories for evaluation. When reporting the generalization log likelihood, as done e.g. in Table 2, we use 10 unseen trajectories. This evaluation is then repeated for 10 different , meaning that we retrain the model 10 times on a data set with the same ground truth, but a different noise realization. We then report the mean and standard deviation of the log likelihood across these repetitions.

**Weight decay**  To prevent overfitting, we use weight decay on the parameters of both the dynamics and the smoother neural networks. We denote by $wd_D$ the weight decay parameter of the dynamics model, and $wd_S$ the weight decay parameters of the smoother model. While we keep the $wd_S$ constant during all three phases of training, we gradually increase $wd_D$ from 0 to its final value, which is the same as $wd_S$. The increase follows a polynomial schedule with power 0.8.

## B.4   Training details

The training of DGM, i.e. optimizing Equation (5), can be split into three distinct phases: *transition*, *training*, and *fine-tuning*. In the *transition* phase we gradually increase the value of both $\lambda$ and the weight decay regularization parameter of the dynamics ($wd_D$) from 0 to its final value. When these parameters reach their final value, we reach the end of the transition phase and start the *training* phase. In this phase, all optimization parameters are left constant. It stops when the last 1000 steps are reached. Then, the *fine-tune* phase starts, where we decrease learning rate to 0.01. The gradual increase of $\lambda$ and $wd_D$ follows polynomial schedule with power 0.8. As an optimizer, we use Adam.

**Supporting points**   The selection of the supporting points in $\mathcal{T}$ is different for data sets consisting of one or multiple trajectories. If there is only one trajectory in the dataset, we match the derivatives at the same places where we observe the state. If there are multiple trajectories, we match the derivatives at 30 equidistant time points on each training trajectory.

**Selection of $\lambda$**   The loss of Equation (5) is a multi-objective optimization problem with a trade-off parameter $\lambda$. Intuitively, if $\lambda$ is too small, the model only tries to fit the data and neglects the dynamics. On the other hand, with too large $\lambda$, the model neglects the data fit and only cares about the dynamics. In Figure 11 we show a plot of log likelihood score on the 10 test trajectories of the LV 100 dataset with varying $\lambda$. We train the model for $\lambda \cdot |\dot{\mathcal{X}}|/|\mathcal{D}| \in \{2^i | i = -20, \ldots, 6\}$. To estimate the robustness of the experiment, we show the mean and standard deviation over 5 different noise realizations.

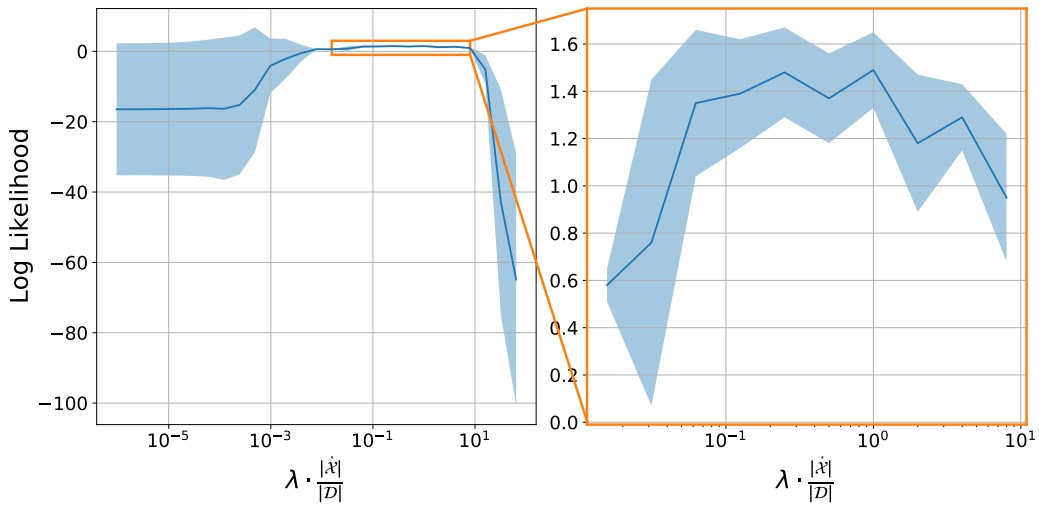

Figure 11: If $\lambda$ is too small, the dynamics model does not regularize the smoother sufficiently, the model overfits to the data and the test log likelihood score is worse. If $\lambda$ is too large, the observation term gets dominated, the model underfits and the log likelihood score on the test data is worse. Empirically, we found that we achieve the best log likelihood on the test data with $\lambda = |\mathcal{D}|/|\dot{\mathcal{X}}|$.

**Parameter selection**   To search for the best performing parameters we performed sweep over learning rate value $lr$ in the transition and training phase and over the weight decay parameters $wd_S$. For $lr$, we considered the values $0.02, 0.05$ and $0.1$. For $wd_S$, we considered $0.1, 0.5$ and $1.0$.

**Training time**   We did not optimize our hyperparameters and code for training time. Nevertheless, we report the length of each phase and the total time in the Table 3.

Table 3: Number of steps for each training phase and total training time for different datasets. For the times, we report mean $\pm$ standard deviation over 10 different random seeds.

|         | Transition | Training | Fine-Tuning | Time[s]        |
|---------|-----------|----------|-------------|----------------|
| LV 1    | 1000      | 0        | 1000        | $329 \pm 15$   |
| LO 1    | 1000      | 0        | 1000        | $399 \pm 6$    |
| DP 1    | 1000      | 1000     | 1000        | $535 \pm 42$   |
| QU 1    | 1000      | 2000     | 1000        | $1121 \pm 41$  |
| LV 100  | 1000      | 0        | 1000        | $408 \pm 8$    |
| LO 125  | 1000      | 0        | 1000        | $753 \pm 8$    |
| DP 100  | 5000      | 4000     | 1000        | $2988 \pm 261$ |
| QU 64   | 6000      | 3000     | 1000        | $8387 \pm 36$  |

## C Bayesian NODE training

In this section, we describe the specifics of the experiments and the implementation of SGLD and SGHMC, the Bayesian integration benchmarks used in the paper. For all experiments, we use a slightly changed version of the code provided by Dandekar et al. (2021), which is written in Julia (Bezanson et al., 2017).

### C.1 Effects of Overparametrization

Here we provide further details regarding the experiment presented in Figure 3. For the ground truth dynamics $\dot{\boldsymbol{x}} = \boldsymbol{A}\boldsymbol{x}$, we selected the matrix $\boldsymbol{A}$ such that it has 1 stable and 2 marginally stable modes. The eigenvalue of the stable mode is selected uniformly at random from $[-0.5, -0.1]$. To create marginally stable modes, we create a block $\boldsymbol{C} \in \mathbb{R}^{2 \times 2}$, where its components are sampled i.i.d. uniformly at random from $[0, 1]$. The marginally stable part is then created as $\boldsymbol{A}$ as $\frac{\pi}{2\rho(\boldsymbol{C} - \boldsymbol{C}^{\top})} \left( \boldsymbol{C} - \boldsymbol{C}^{\top} \right)$, where $\rho(.)$ denotes the spectral radius. Using the spectral radius in the normalization ensures that the period of the marginally stable mode is bounded with $\pi/2$. We selected the initial condition for the trajectory uniformly at random from the unit sphere in $\mathbb{R}^3$. We evolved the trajectory on the time interval $(0, 10)$ and observed 100 noisy observations, where every ground truth value was perturbed with additive, independent Gaussian noise, drawn from $\mathcal{N}\left(0, 0.1^2\right)$.

While DGM performed without any pre-training, we observed that both SGLD and SGHMC struggle without narrow priors. To obtain reasonable priors, we followed the following procedure:

First, we pretrain every set of matrices $\boldsymbol{B}_1, \ldots, \boldsymbol{B}_k$ on the ground truth, such that the

$$\sum_{i=1}^{100} \left\| \dot{\boldsymbol{x}}(t_i) - \prod_{j=1}^{k} \boldsymbol{B}_j \boldsymbol{x}(t_i) \right\|_2^2 \leq 10^{-5},$$

where $t_i$ are times at which we observed the state. We selected the prior as Gaussian centered around the pretrained parameters. The standard deviation was chosen such that the standard deviation of the product of the matrices $\prod_{j=1}^{k} \boldsymbol{B}_j$ stays at 0.01, independent of the number of matrices used. For SGHMC, we selected learning rate $1.5 \times 10^{-7}$ and momentum decay 0.1 as hyperparameters. For SGLD, we chose the hyperparameters $a = 0.001, b = 1$ and $\gamma = 0.9$, which SGLD then uses to calculate a polynomial decay $a(b + k)^{-\gamma}$ (at step $k$) for its learning rate. For sampling we used 5 chains with 20000 samples on each chain. The last 2000 samples of each chain were used for evaluation. With this setting we ensured that the $r$-hat score was smaller than 1.1.

### C.2 Finetuning for Benchmark Systems

Both SGLD and SGHMC require hyperparameters, that influence their performance quite strongly. In this subsection, we explain all the tuning steps and requirements we deployed for these algorithms to produce the results shown in the main paper. For both algorithms, we used a setup of 6 chains, that were sampled in parallel, with 10000 samples per chain, where the final 2000 samples were taken as predictions. These numbers were selected using the r-hat value to determine if the chains had sufficiently converged.

**Hyperparameters of** SGLD  For SGLD, we additionally searched over the hyperparameters $a$, $b$, and $\gamma$. All three are used to calculate the learning rate of the algorithm. These parameters were chosen by evaluating the log likelihood of the ground truth on a grid, which was given by the values $a \in \{0.0001, 0.001, 0.005, 0.01, 0.05, 0.1\}$, $b \in \{0.3, 0.6, 1.0, 1.5, 2\}$ and $\gamma \in \{0.5001, 0.55, 0.6, 0.7, 0.8, 0.99\}$. Clearly, using the log likelihood of the ground truth to tune the hyperparameters overestimates the performance of the algorithm, but it provides us with an optimistic estimate of its real performance in practice. For computational reasons, we only performed a grid search on the one trajectory experiments, and then reused the same hyperparameters on the multi-trajectory experiments. All hyperparameters are shown in Table 4.

**Hyperparameters of** SGHMC  For SGHMC, we additionally searched over the hyperparameters the learning rate and the momentum decay. Again, these parameters were chosen by evaluating

the log likelihood of the ground truth on a grid, where learning rate was chosen from the set $\{1e{-}8, 5e{-}8, 1.5e{-}7, 5e{-}7, 1e{-}6, 5e{-}6, 1e{-}5, 5e{-}5, 1e{-}4, 5e{-}4\}$ and momentum decay was chosen from the set $\{0.0001, 0.001, 0.05, 0.1, 0.5, 1, 5\}$. Since we used the log likelihood of the ground truth again to tune the hyperparameters, we overestimate the performance of the algorithm and obtain thus an optimistic estimate of its real performance in practice. For computational reasons, be only performed a grid search on the one trajectory experiments, and then reused the same hyperparameters on the multi-trajectory experiments. All hyperparameters are shown in Table 4.

Table 4: Hyperparameters with best performance evaluated on the likelihood of the ground truth. The hyperparameters are different for the parametric (p) and the non-parametric (n) dynamics models, which is indicated with the last letter.

| | SGLD | | | SGHMC | |
|---|---|---|---|---|---|
| | $a$ | $b$ | $\gamma$ | learning rate | momentum decay |
| Lotka Volterra p | 1e−3 | 2 | 0.5001 | 5e−7 | 0.1 |
| Lorenz p | 0.001 | 1.5 | 0.5001 | 1e−5 | 0.05 |
| Double Pendulum p | 0.1 | 0.3 | 0.5001 | 1e−6 | 0.1 |
| Quadrocopter p | 0.0001 | 1.5 | 0.7 | 5e−7 | 0.5 |
| Lotka Volterra n | 0.005 | 1.5 | 0.7 | 5e−7 | 0.5 |
| Lorenz n | 0.001 | 1.5 | 0.55 | 5e−6 | 0.1 |
| Double Pendulum n | 0.01 | 2 | 0.55 | 1e−6 | 0.05 |
| Quadrocopter n | F | F | F | 5e−7 | 0.05 |

**Choice of Priors for** SGLD **and** SGHMC    Since SGLD and SGHMC are both Bayesian methods, they need to be supplied with a prior. As we discovered in our experiments, this prior plays a crucial role in the stability of the algorithm. In the end, we did not manage to get them to converge without some use of ground truth. In particular, if the priors were not chosen narrowly around some ground truth, the algorithms just returned NaNs, since their integration scheme runs into numerical issues. For the parametric case shown in Appendix E, where we assume access to the true parametric form of the system, we thus chose narrow priors around the ground truth of the parameter values, that were used to create the data set. For Lotka Volterra, we chose a uniform distribution around the ground truth $\pm 0.5$, i.e. $\theta_i \sim \text{Uniform}[0.5, 1]$ for all components of $\boldsymbol{\theta}$. For Lorenz, we chose $\alpha \sim \text{Uniform}[8, 12]$, $\beta \sim \text{Uniform}[25, 31]$ and $\gamma \sim \text{Uniform}[6/3, 10/3]$. For Double Pendulum, we chose $m \sim \text{Uniform}[0.5, 1.5]$ and $l \sim \text{Uniform}[0.5, 1.5]$. For Quadrocopter, we chose independent Gaussians, centered around the ground truth, with a standard deviation of $0.005$. For all experiments, the prior on the observation noise was set to $\sigma \sim \text{InverseGamma}[2, 3]$, except for SGLD when inferring the Lorenz system. There, we had to fix the noise standard deviation to its ground truth, to get convergence.

For the non-parametric case shown in the main paper, we needed a different strategy, since no ground truth information was available for the weights of the neural dynamics model. Thus, we first trained a deterministic dynamics model. As data, we sampled 100 tuples $\boldsymbol{x}, \dot{\boldsymbol{x}}$ equidistantly in time on the trajectory. Note that we implicitly assume access to the ground truth of the dynamics model, i.e. we assume we are provided with accurate, noise free $\dot{\boldsymbol{x}}$. The neural dynamics model was then pre-trained on these pairs, until the loss was almost zero (up to $1e{-}5$). SGLD and SGHMC were then provided with Gaussian priors, independent for each component of $\boldsymbol{\theta}$, centered around the pre-trained weights, with a standard deviation of 0.1.

### C.3    Number of integrations for prediction

SGLD and SGHMC both return samples of the parameters of the dynamics model. To obtain uncertainties in the state space at prediction time, each one of these samples needs to be turned into a sample trajectory, by using numerical integration. To obtain maximum accuracy, we would ideally integrate all parameter samples obtained by the chains. However, due to the computational burden inflicted by numerical integration, this is not feasible. We thus need to find a trade-off between accuracy and computational cost, by randomly subsampling the number of available parameter samples.

In Figure 12 we show how the log likelihood of the ground truth changes with increasing number of sample trajectories on the LV 1 dataset. After initial fluctuations, the log likelihood of the ground truth stabilizes after approximately 200 steps. To obtain the results of Table 1, we thus chose 200 integration steps.

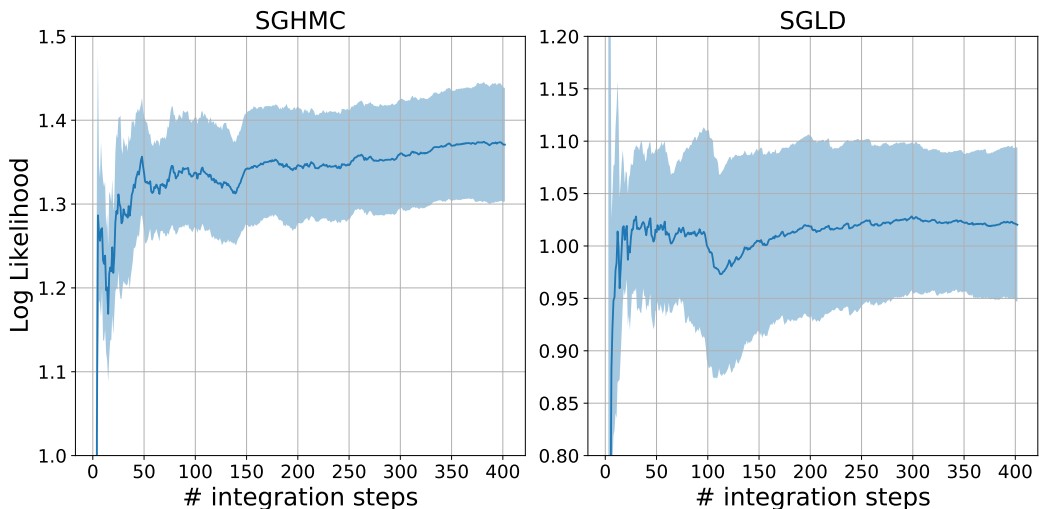

Figure 12: We select the number of sample trajectories for uncertainty prediction to be 200, since we observe that the log likelihood of the ground truth stops fluctuating after 200 steps.

# D   Scaling to many observations or trajectories

Let $N$ be the total number of observations, summed over all training trajectories. In this section, we will analyze the computational complexity of DGM in terms of $N$ and demonstrate how this can be drastically reduced using standard methods from the literature. For notational compactness, we will assume that the supporting points in $\mathcal{T}$ are at the same locations as the observations in $\mathcal{D}$. However, this is by no means necessary. As long as they are chosen to be constant or stand in a linear relationship to the number of observations, our analysis still holds. We will thus use $\boldsymbol{x}$ and $\boldsymbol{x}_{\text{supp}}$ and the corresponding quantities interchangeably. Similarly, we will omit the $k$ that was used for indexing the state dimension and assume one-dimensional systems. The extension to multi-dimensional systems is straight forward and comes at the cost of an additional factor $K$.

Fortunately, most components of the loss of DGM given by Equation (5) can be calculated in linear time. In particular, it is worth noting that the independence assumption made when calculating the Wasserstein distance in Equation (16) alleviates the need to work with the full covariance matrix and lets us work with its diagonal elements instead. Nevertheless, there are several terms that are not straight forward to calculate. Besides the marginal log likelihood of the observations, these are the posteriors

$$p_S(\boldsymbol{x}|\mathcal{D}, \mathcal{T}) = \mathcal{N}\left(\boldsymbol{x}|\boldsymbol{\mu}_{\text{post}}, \boldsymbol{\Sigma}_{\text{post}}\right), \tag{17}$$

$$p_S(\dot{\mathcal{X}}|\mathcal{D}, \mathcal{T}) = \mathcal{N}\left(\dot{\boldsymbol{x}}_{\text{supp}}|\boldsymbol{\mu}_S, \boldsymbol{\Sigma}_S\right), \tag{18}$$

where

$$\boldsymbol{\mu}_{\text{post}} = \boldsymbol{\mu} + \mathcal{K}^T(\mathcal{K} + \sigma^2\boldsymbol{I})^{-1}(\boldsymbol{y} - \boldsymbol{\mu}), \tag{19}$$

$$\boldsymbol{\Sigma}_{\text{post}} = \mathcal{K} - \mathcal{K}^T(\mathcal{K} + \sigma^2\boldsymbol{I})^{-1}\mathcal{K}, \tag{20}$$

$$\boldsymbol{\mu}_S = \dot{\boldsymbol{\mu}} + \dot{\mathcal{K}}(\mathcal{K} + \sigma^2\boldsymbol{I})^{-1}(\boldsymbol{y} - \boldsymbol{\mu}), \tag{21}$$

$$\boldsymbol{\Sigma}_S = \ddot{\mathcal{K}} - \dot{\mathcal{K}}(\mathcal{K} + \sigma^2\boldsymbol{I})^{-1}\dot{\mathcal{K}}^{\top}. \tag{22}$$

Here, Equation (17) is used for prediction, while its mean is also used in the approximation of Equation (16). On the other hand, Equation (18) is used directly for Equation (16). Note that in both cases, we only need the diagonal elements of the covariance matrices, a fact that will become important later on.

In its original form, calculating the matrix inverses of both Equation (17) and Equation (18) has cubic complexity in $N$. To alleviate this problem, we follow Rahimi et al. (2007) and Angelis et al. (2020) by using a feature approximation of the kernel matrix and its derivatives. In particular, let $\boldsymbol{\Phi} \in \mathbb{R}^{F \times N}$ be a matrix of $F$ random Fourier features as described by Rahimi et al. (2007). Furthermore, denote $\dot{\boldsymbol{\Phi}}$ as its derivative w.r.t. the time input variable, as defined by Angelis et al. (2020). We can now approximate the kernel matrix and its derivative versions as

$$\boldsymbol{K} \approx \boldsymbol{\Phi}^{\top}\boldsymbol{\Phi}, \quad \dot{\mathcal{K}}^{\top} \approx \dot{\boldsymbol{\Phi}}^{\top}\boldsymbol{\Phi}, \quad \text{and} \quad \ddot{\mathcal{K}} \approx \dot{\boldsymbol{\Phi}}^{\top}\dot{\boldsymbol{\Phi}}. \tag{23}$$

Using these approximations, we can leverage the Woodbury idendity to approximate

$$(\mathcal{K} + \sigma^2\boldsymbol{I})^{-1} \approx \frac{1}{\sigma^2}\left[\boldsymbol{I} - \boldsymbol{\Phi}^{\top}\left(\boldsymbol{\Phi}\boldsymbol{\Phi}^{\top} + \sigma^2\boldsymbol{I}\right)^{-1}\boldsymbol{\Phi}\right]. \tag{24}$$

This approximation allows us to invert a $F \times F$ matrix, to replace the inversion of a $N \times N$ matrix. This can be leveraged to calculate

$$\boldsymbol{\mu}_S = \dot{\boldsymbol{\mu}} + \dot{\mathcal{K}}(\mathcal{K} + \sigma^2\boldsymbol{I})^{-1}(\boldsymbol{y} - \boldsymbol{\mu}) \tag{25}$$

$$\approx \dot{\boldsymbol{\mu}} + \frac{1}{\sigma^2}\dot{\boldsymbol{\Phi}}^{\top}\boldsymbol{\Phi}\left[\boldsymbol{I} - \boldsymbol{\Phi}^{\top}\left(\boldsymbol{\Phi}\boldsymbol{\Phi}^{\top} + \sigma^2\boldsymbol{I}\right)^{-1}\boldsymbol{\Phi}\right](\boldsymbol{y} - \boldsymbol{\mu}) \tag{26}$$

and

$$\boldsymbol{\Sigma}_S = \ddot{\mathcal{K}} - \dot{\mathcal{K}}(\mathcal{K} + \sigma^2\boldsymbol{I})^{-1}\dot{\mathcal{K}}^{\top} \tag{27}$$

$$\approx \dot{\boldsymbol{\Phi}}^{\top}\dot{\boldsymbol{\Phi}} - \frac{1}{\sigma^2}\dot{\boldsymbol{\Phi}}^{\top}\boldsymbol{\Phi}\left[\boldsymbol{I} - \boldsymbol{\Phi}^{\top}\left(\boldsymbol{\Phi}\boldsymbol{\Phi}^{\top} + \sigma^2\boldsymbol{I}\right)^{-1}\boldsymbol{\Phi}\right]\boldsymbol{\Phi}^{\top}\dot{\boldsymbol{\Phi}}. \tag{28}$$

Evaluating the matrix multiplications of Equation (26) in the right order leads to a computational complexity of $\mathcal{O}(NF^2 + F^3)$. Similarly, the diagonal elements of the covariance given by Equation (18) can be calculated with the same complexity, by carefully summarizing everything in between $\dot{\mathbf{\Phi}}^\top$ and $\dot{\mathbf{\Phi}}$ as one $F \times F$ matrix and then calculating the $N$ products independently.

Since the components of Equation (17) have the exact same form as the components of Equation (18), they can be approximated in the exact same way to obtain the exact same computational complexity. Thus, the only components that need further analysis are the components of the marginal log likelihood of the observations, particularly

$$\boldsymbol{y}^\top (\mathcal{K} + \sigma^2 \boldsymbol{I})^{-1} \boldsymbol{y} \approx \boldsymbol{y}^\top \frac{1}{\sigma^2} \left[ \boldsymbol{I} - \mathbf{\Phi}^\top \left( \mathbf{\Phi}\mathbf{\Phi}^\top + \sigma^2 \boldsymbol{I} \right)^{-1} \mathbf{\Phi} \right] \boldsymbol{y} \tag{29}$$

and

$$\mathrm{logdet}(\mathcal{K} + \sigma^2 \boldsymbol{I}) \approx \mathrm{logdet}(\mathbf{\Phi}^\top \mathbf{\Phi} + \sigma^2 \boldsymbol{I}) \tag{30}$$

$$\approx \mathrm{logdet}(\mathbf{\Phi}\mathbf{\Phi}^\top + \sigma^2 \boldsymbol{I}) + (N - F)\mathrm{log}(\sigma^2). \tag{31}$$

In the last line, we used the fact that the nonzero eigenvalues of the transposed of a matrix stay the same.

Combining all these tricks, it is clear that the overall complexity of DGM can be reduced to $\mathcal{O}(NF^2 + F^3)$. Since $F$ is a constant controlling the quality of the approximation scheme and is usually chosen to be constant, we thus get essentially linear computational complexity in the number of observations. Note that these derivations are completely independent of what scheme is chosen to obtain the feature matrix $\mathbf{\Phi}$. For ease of implementation, we opted for random Fourier features though in our experiments.

**Experimental Proof of Concept** To demonstrate that this approximation scheme can be used in the context of DGM, we tested it on the multi-trajectory experiment of Lotka Volterra. To this end, we increased the grid from 10 points per dimension to 25, leading to a total number of 3125 observations instead of 500. As an approximation, we used 50 random Fourier features. Through this approximation, DGM became slightly more sensitive to the optimization hyperparameters. Nevertheless, it reached comparable accuracy within roughly 440 seconds of training, compared to the 408 seconds needed to train the approximation free version on LV 100.

# E   Additional experiments

In this section, we first show the state predictions of DGM on the datasets with multiple trajectories. Then, we compare DGM with SGLD and SGHMC for the parametric case, i.e. where we assume to have access to the true parametric form of the dynamics. Since most datasets have too many trajectories to be fully visualized, we show a random subset instead.

## E.1   Sample plots from trained trajectories

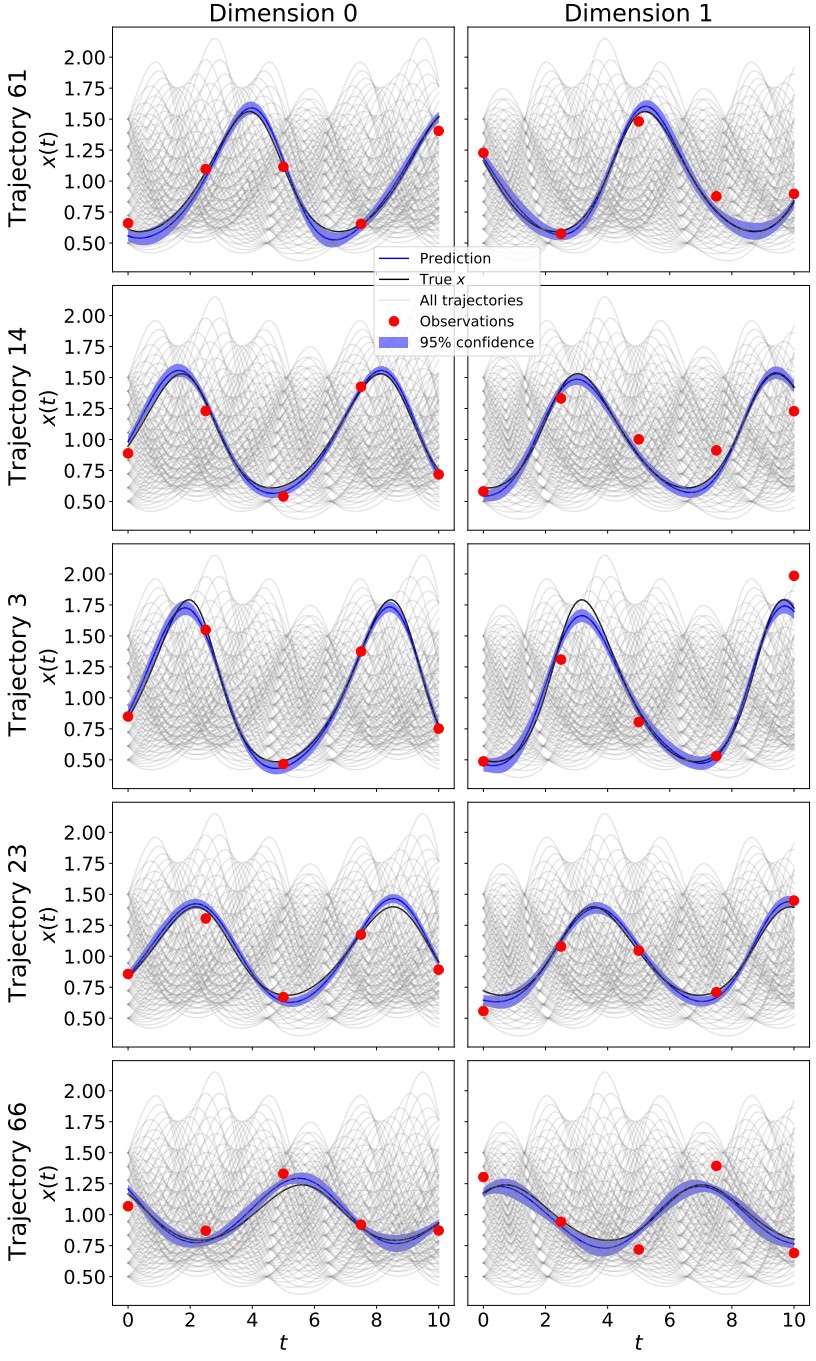

Figure 13: DGM's prediction on 5 randomly sampled training trajectories of LV 100.

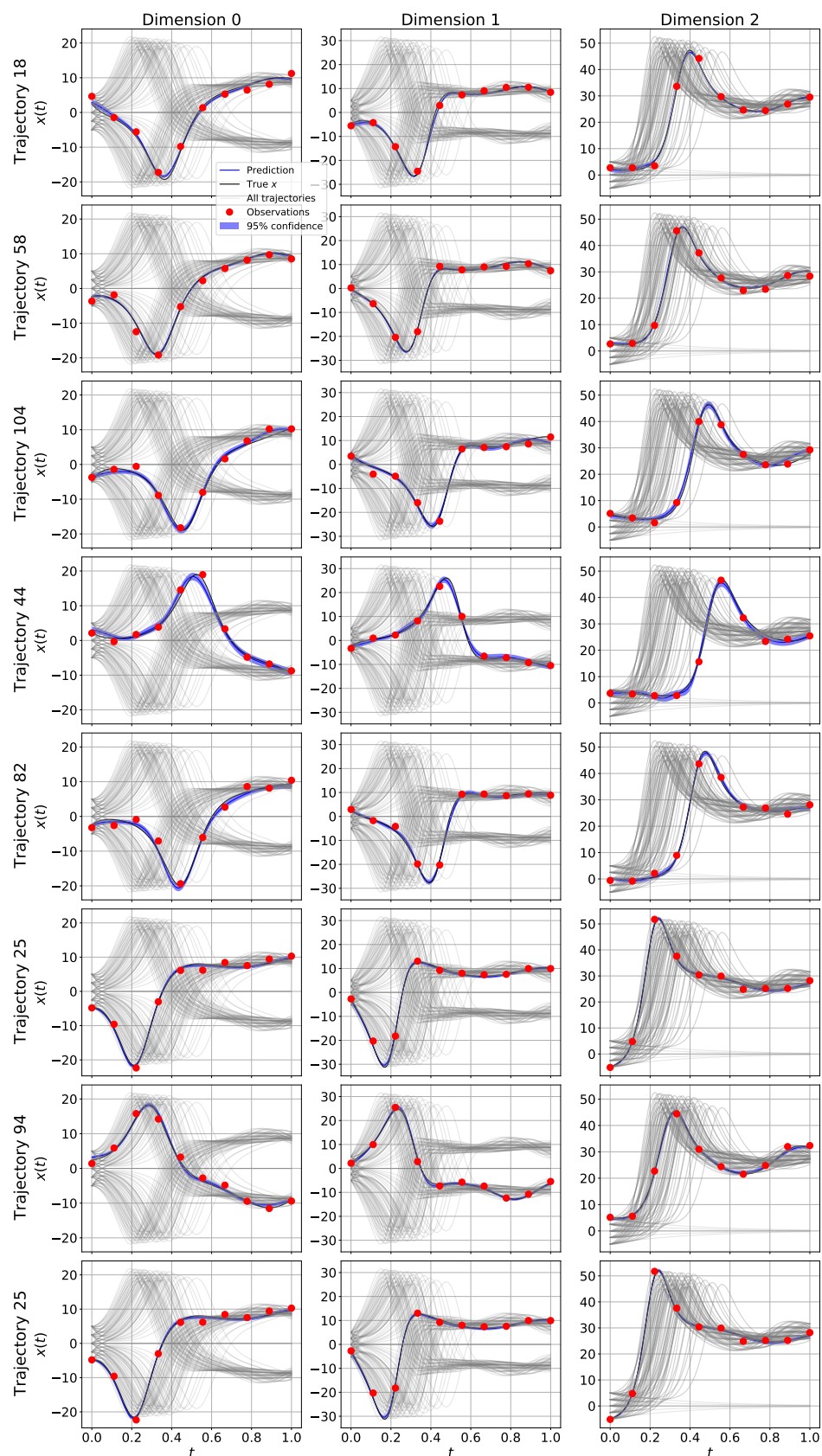

Figure 14: DGM's prediction on 8 randomly sampled training trajectories of LO 125.

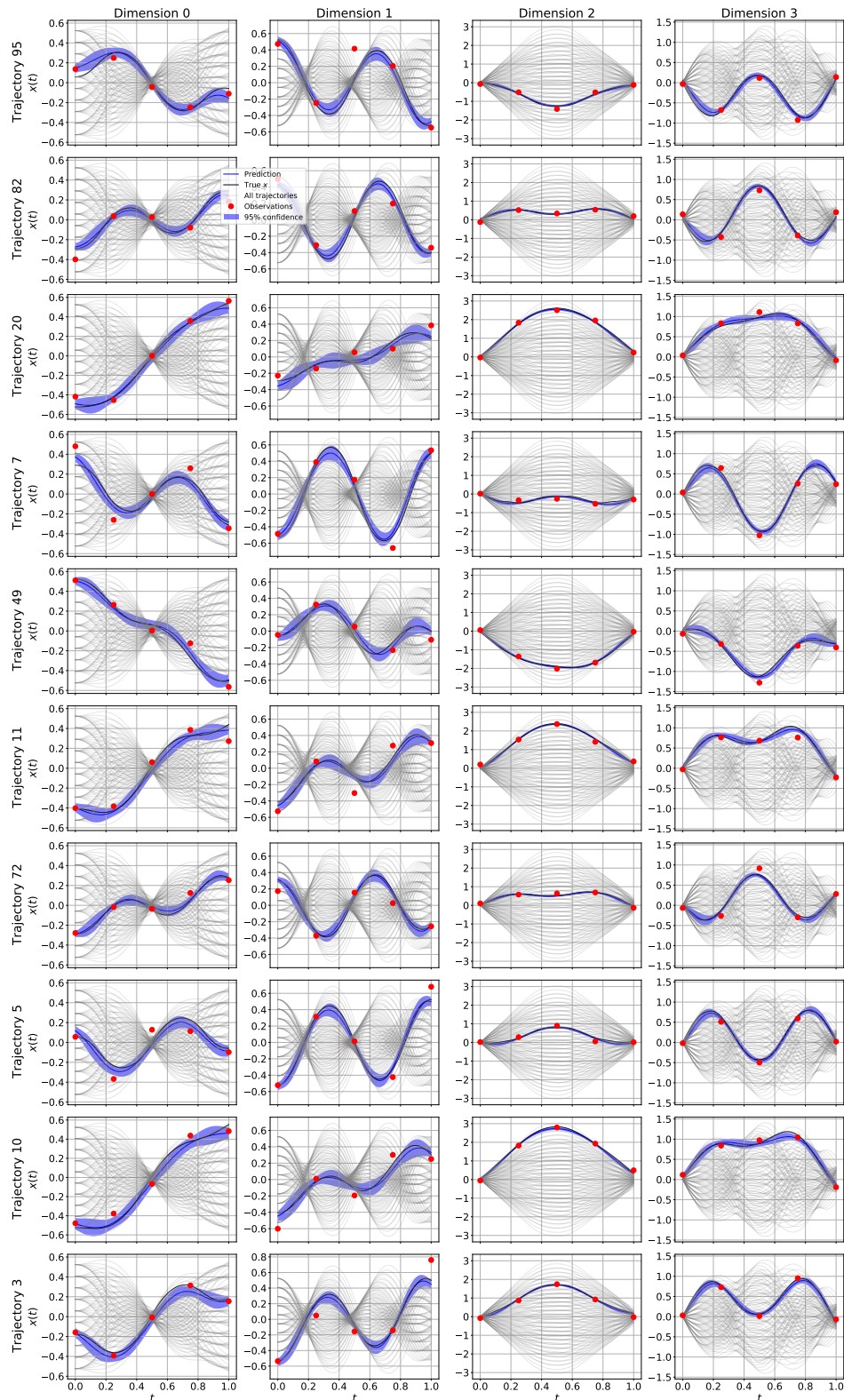

Figure 15: DGM's prediction on 10 randomly sampled training trajectories of DP 100.

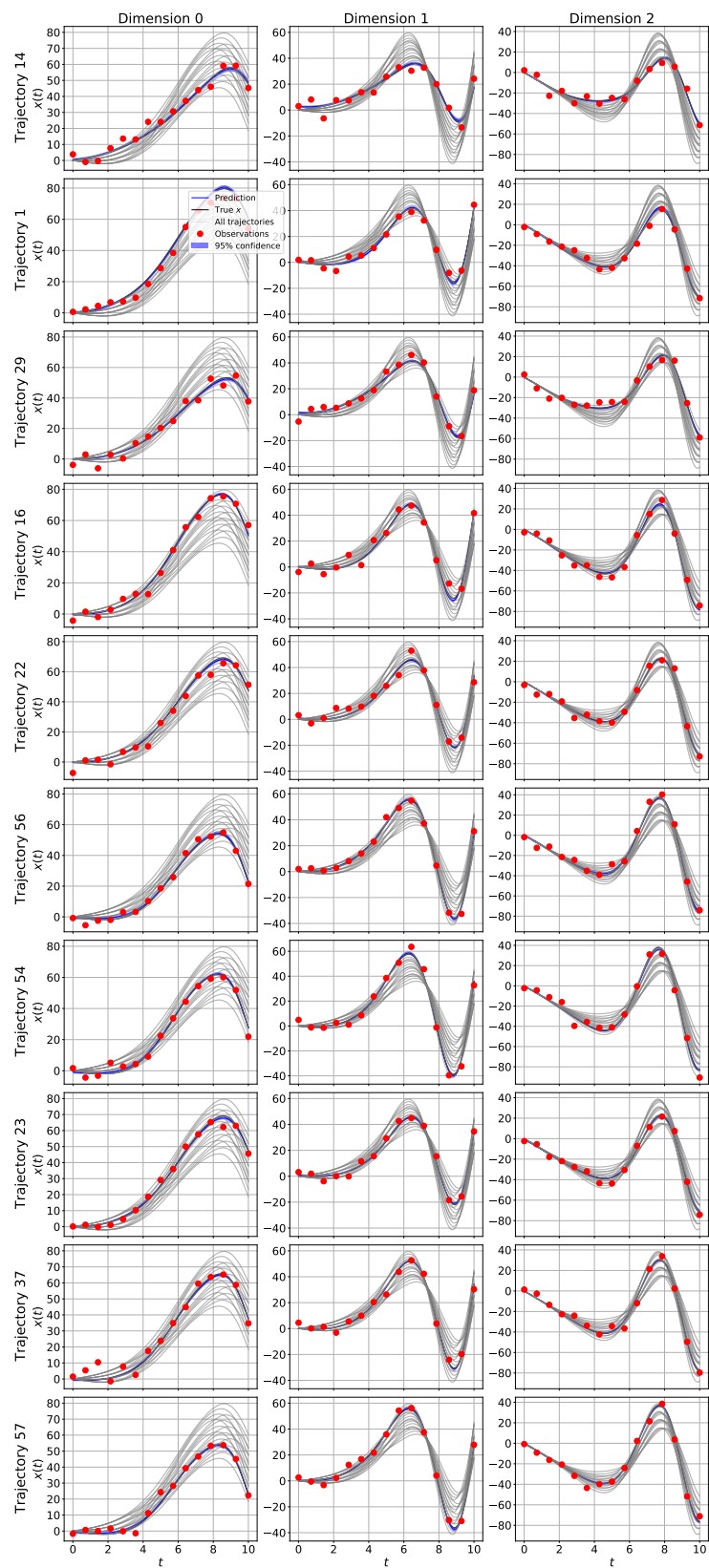

Figure 16: DGM's prediction on 10 randomly sampled training trajectories of QU 64, for state dimensions 0-2.

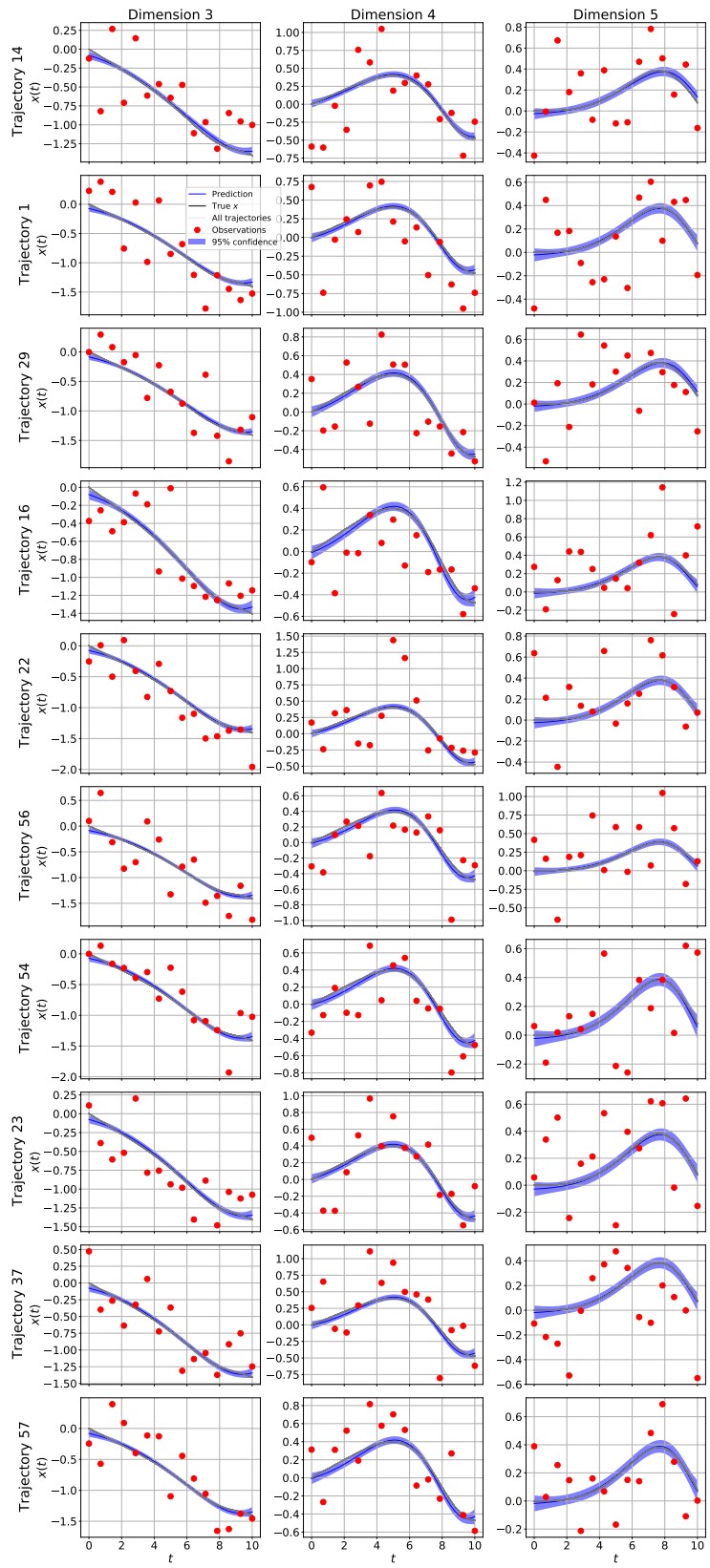

Figure 17: DGM's prediction on 10 randomly sampled training trajectories of QU 64, for state dimensions 3-5.

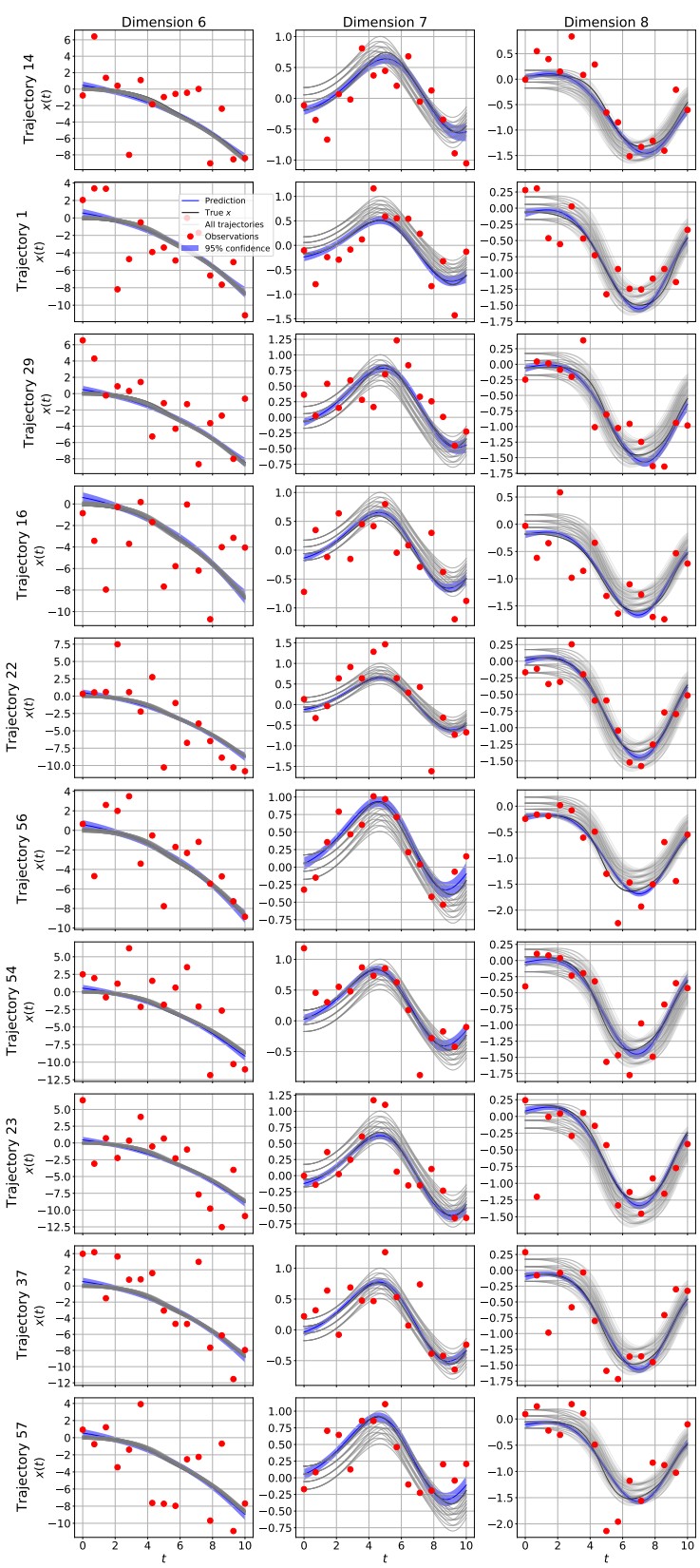

Figure 18: DGM's prediction on 10 randomly sampled training trajectories of QU 64, for state dimensions 6-8.

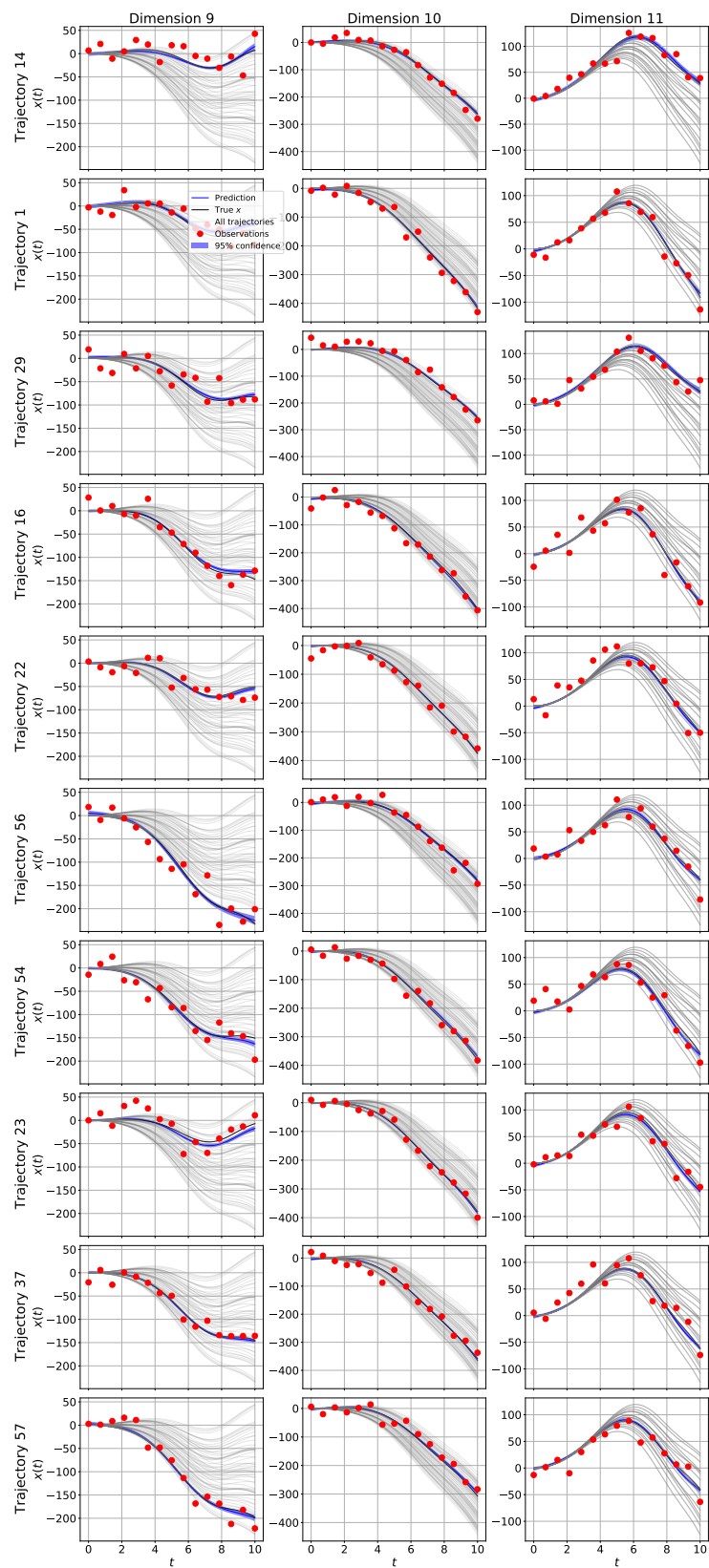

Figure 19: DGM's prediction on 10 randomly sampled training trajectories of QU 64, for state dimensions 9-11.

## E.2  Sample plots from test trajectories

Here, we show DGM's predictions on the test trajectories used to test generalization, as introduced in Appendix A. Since LV 100 is a two dimensional system, we also show the placement of the train and test initial conditions in Figure 20.

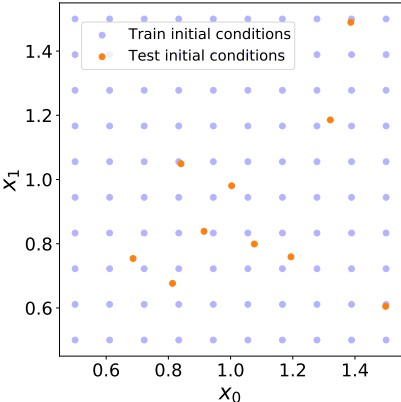

Figure 20: Placement of the initial conditions for the train and test trajectories of the LV 100 dataset. We selected the initial conditions for the train trajectories by gridding $[0.5, 1.5]^2$ with 10 points in every dimension. We select initial conditions for test trajectories independently, uniformly at random from the cube $[0.5, 1.5]^2$.

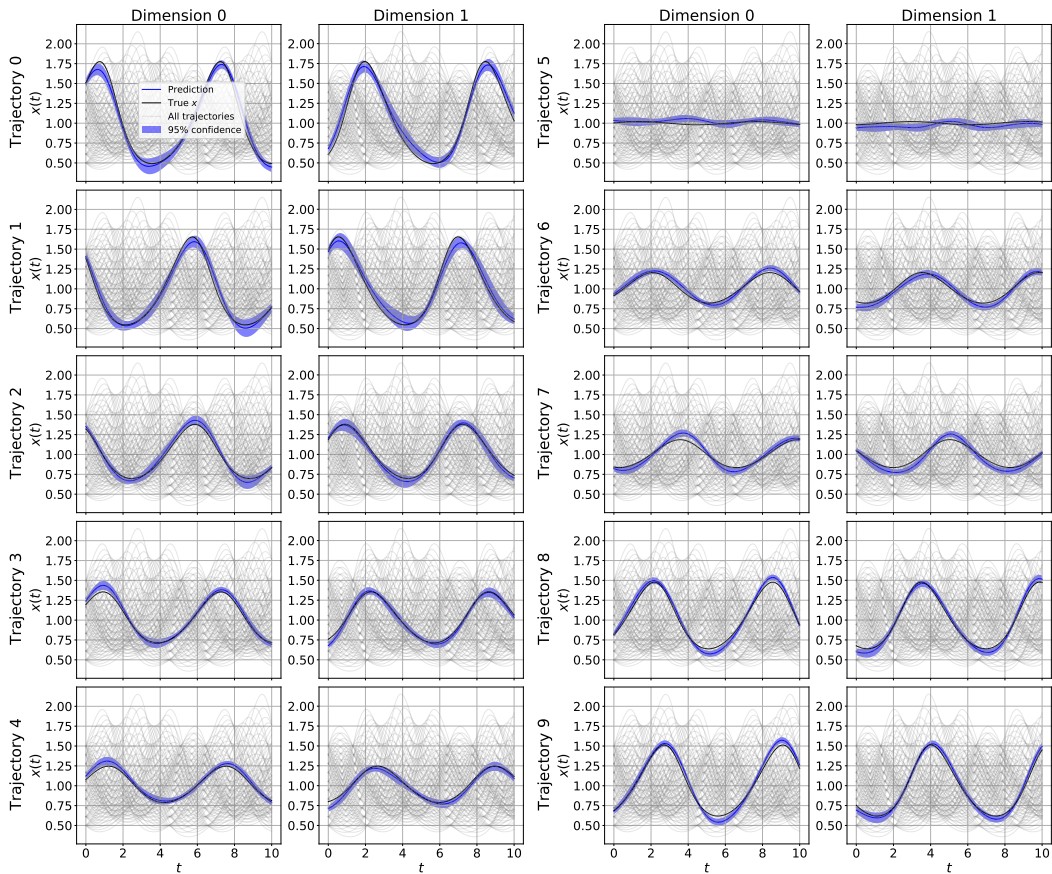

Figure 21: DGM's prediction on 10 randomly sampled test trajectories for the LV 100 dataset.

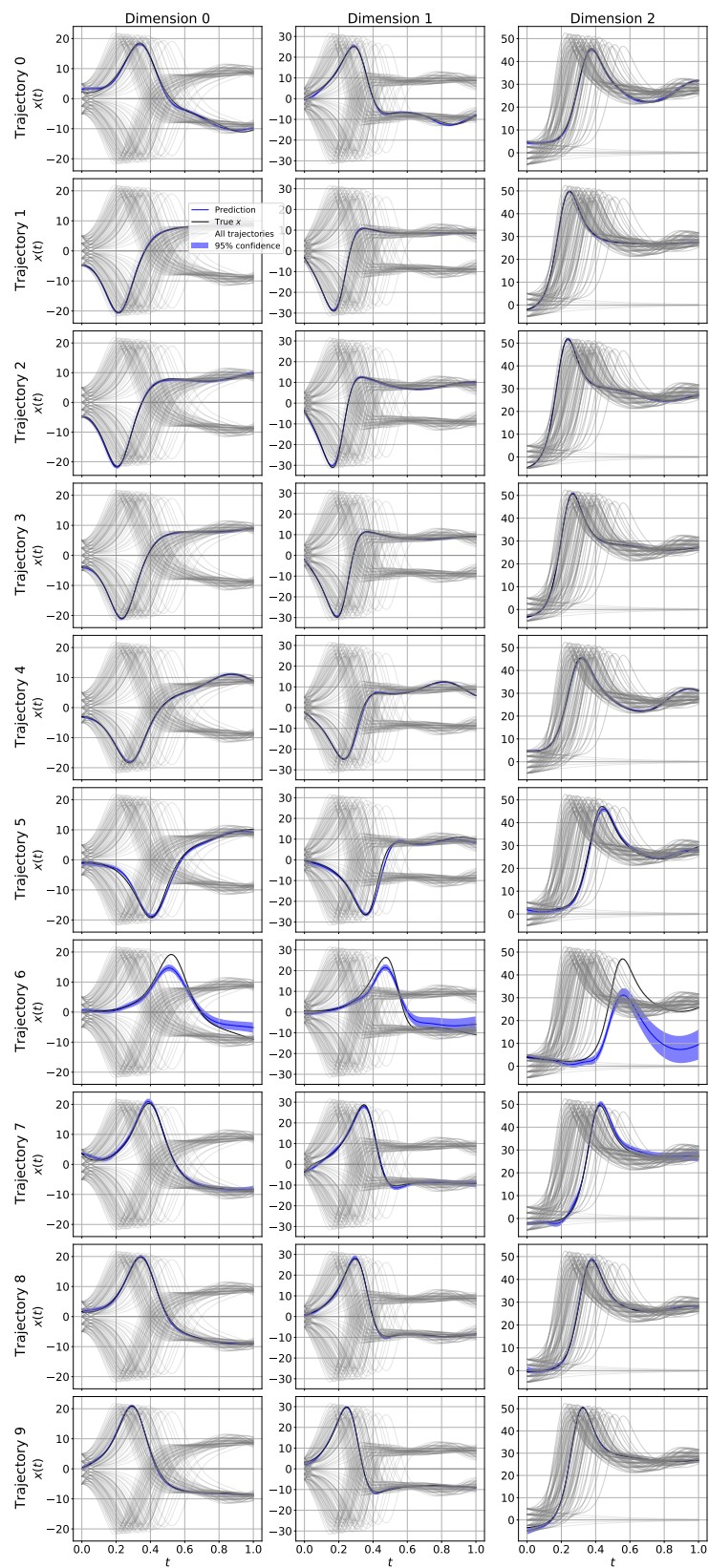

Figure 22: DGM's prediction on 10 randomly sampled test trajectories for the LO 125 dataset.

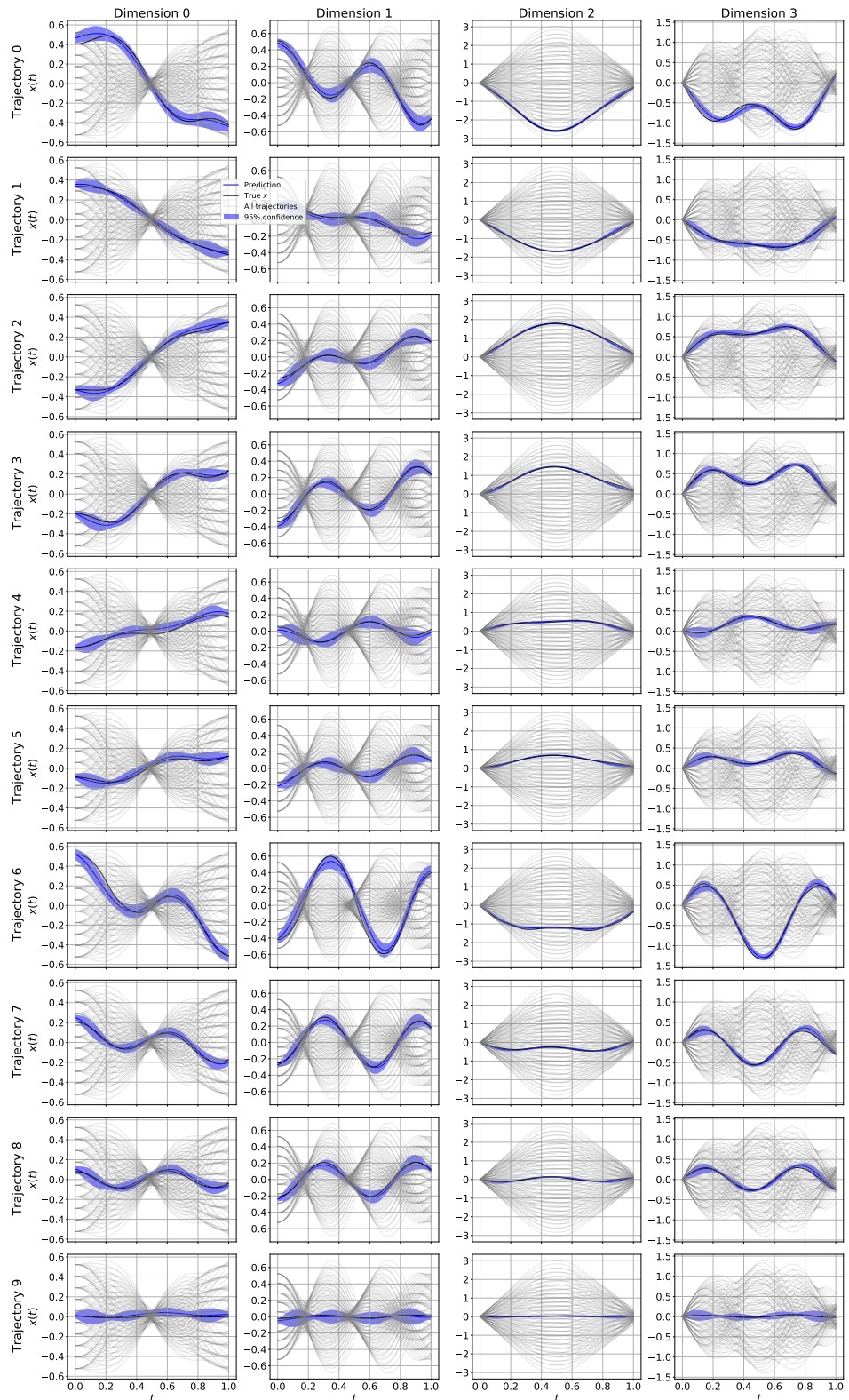

Figure 23: DGM's prediction on 10 randomly sampled test trajectories for the DP 100 dataset.

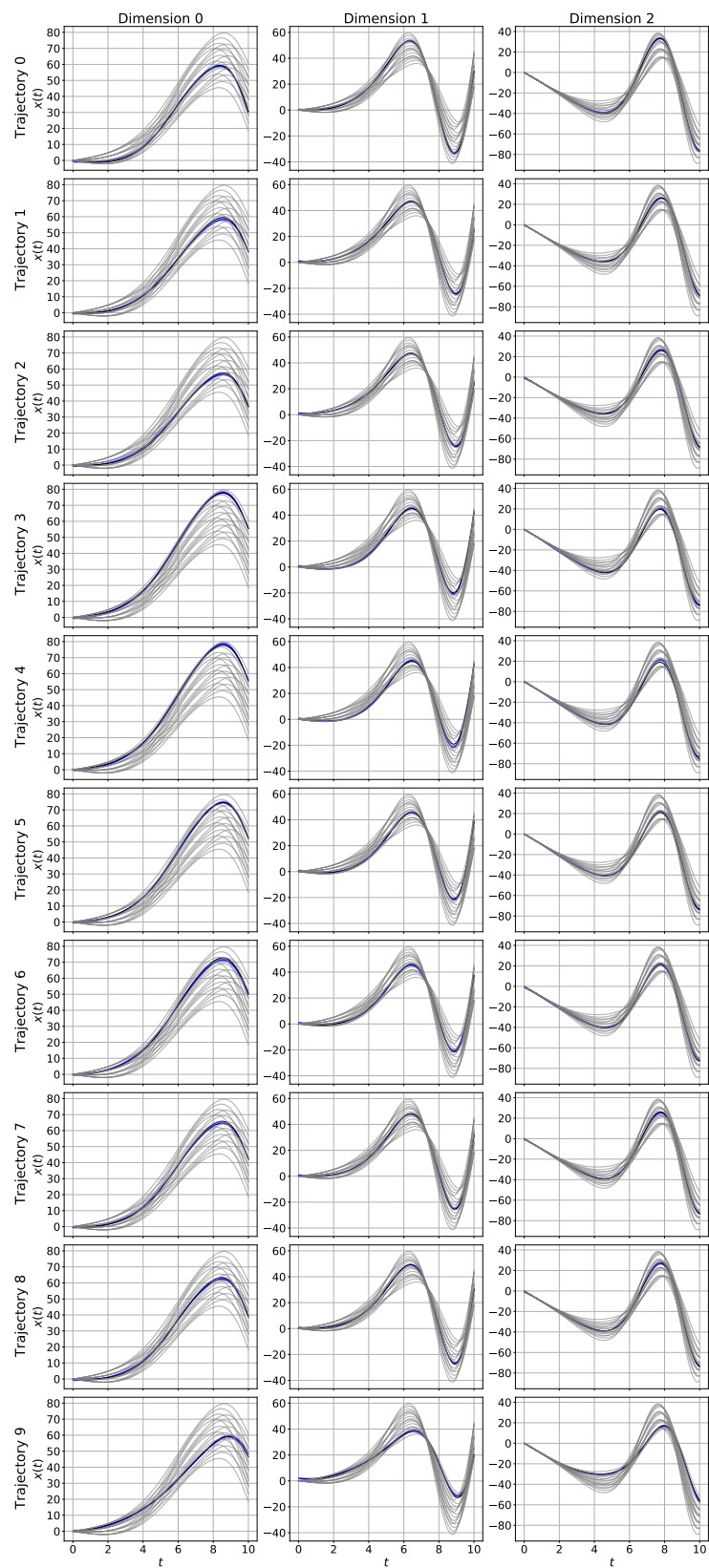

Figure 24: DGM's prediction on 10 randomly sampled test trajectories of QU 64, for state dimensions 0-2.

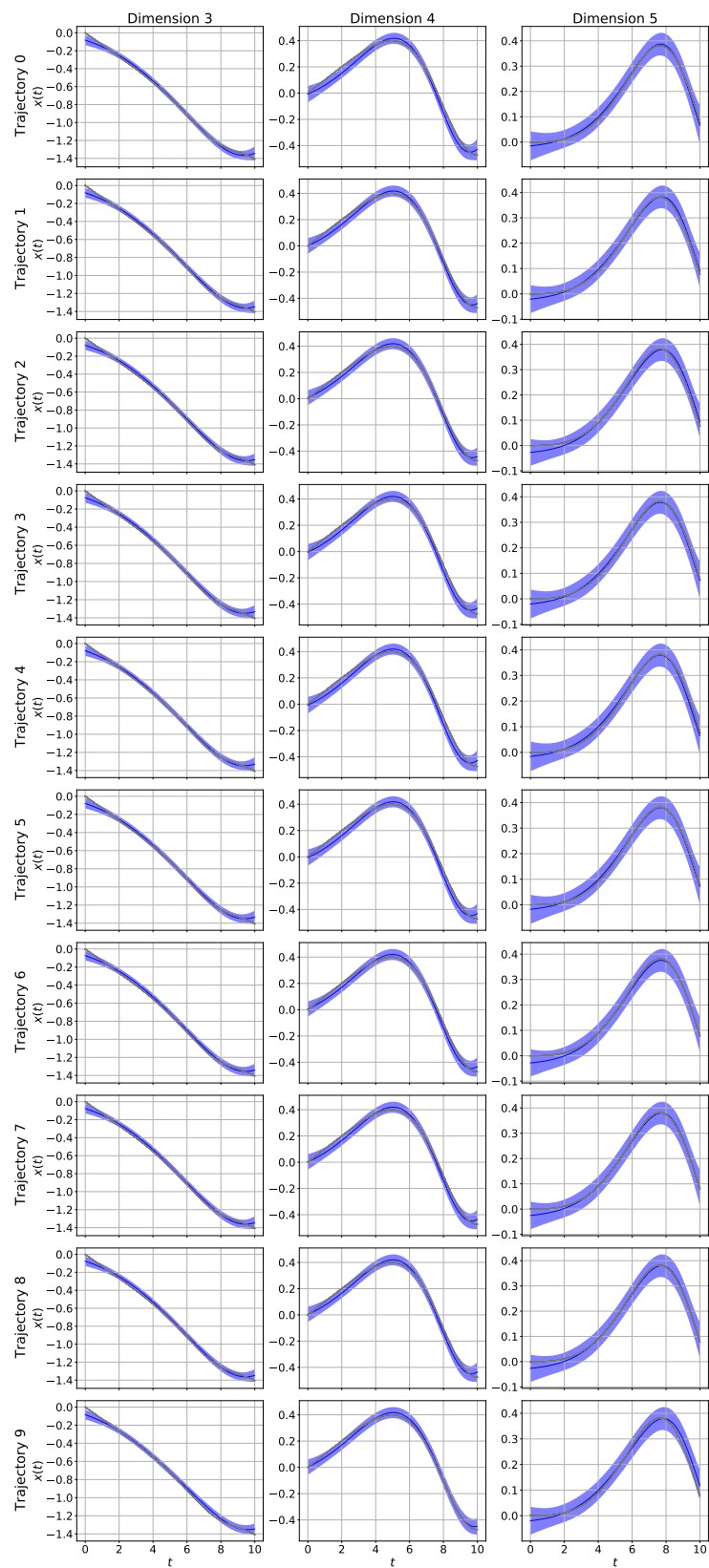

Figure 25: DGM's prediction on 10 randomly sampled test trajectories of QU 64, for state dimensions 3-5.

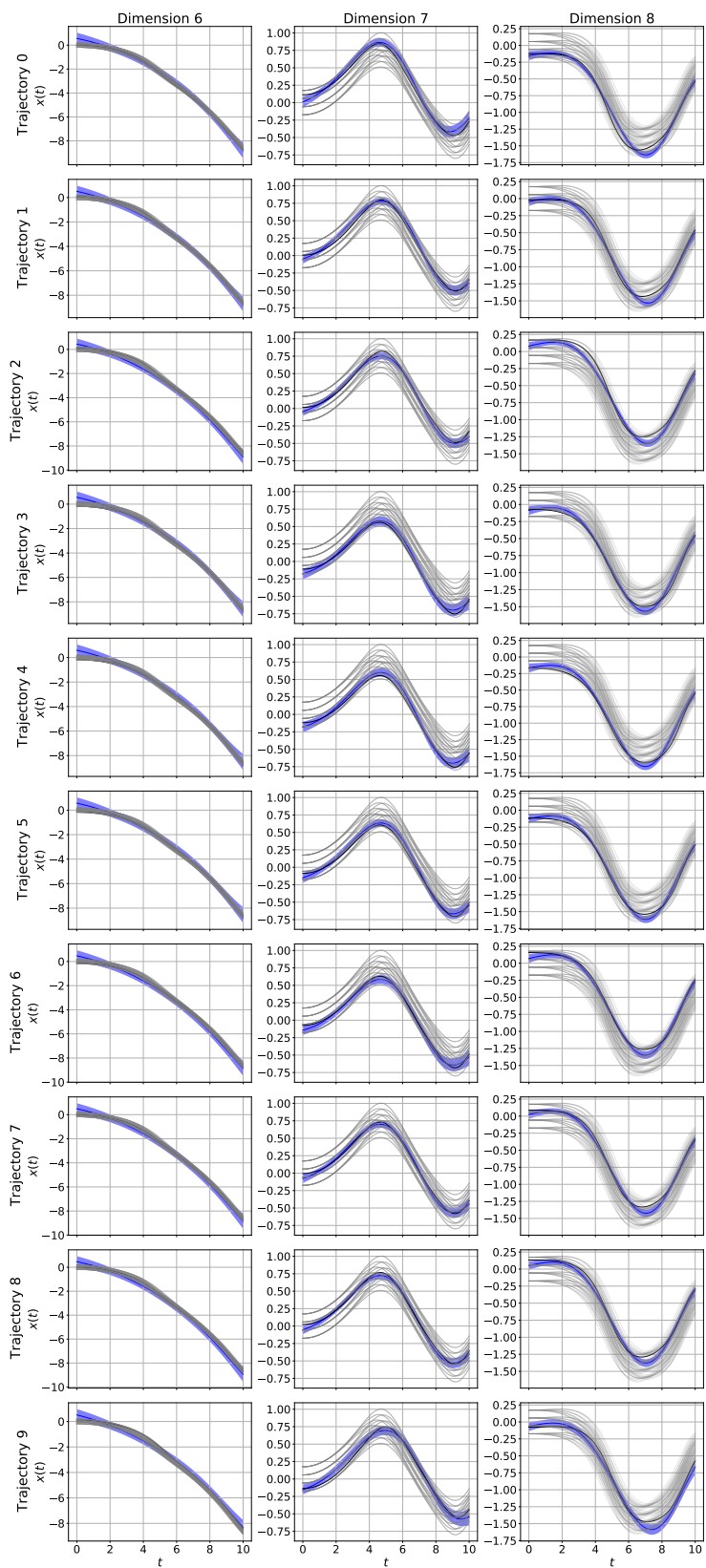

Figure 26: DGM's prediction on 10 randomly sampled test trajectories of QU 64, for state dimensions 6-8.

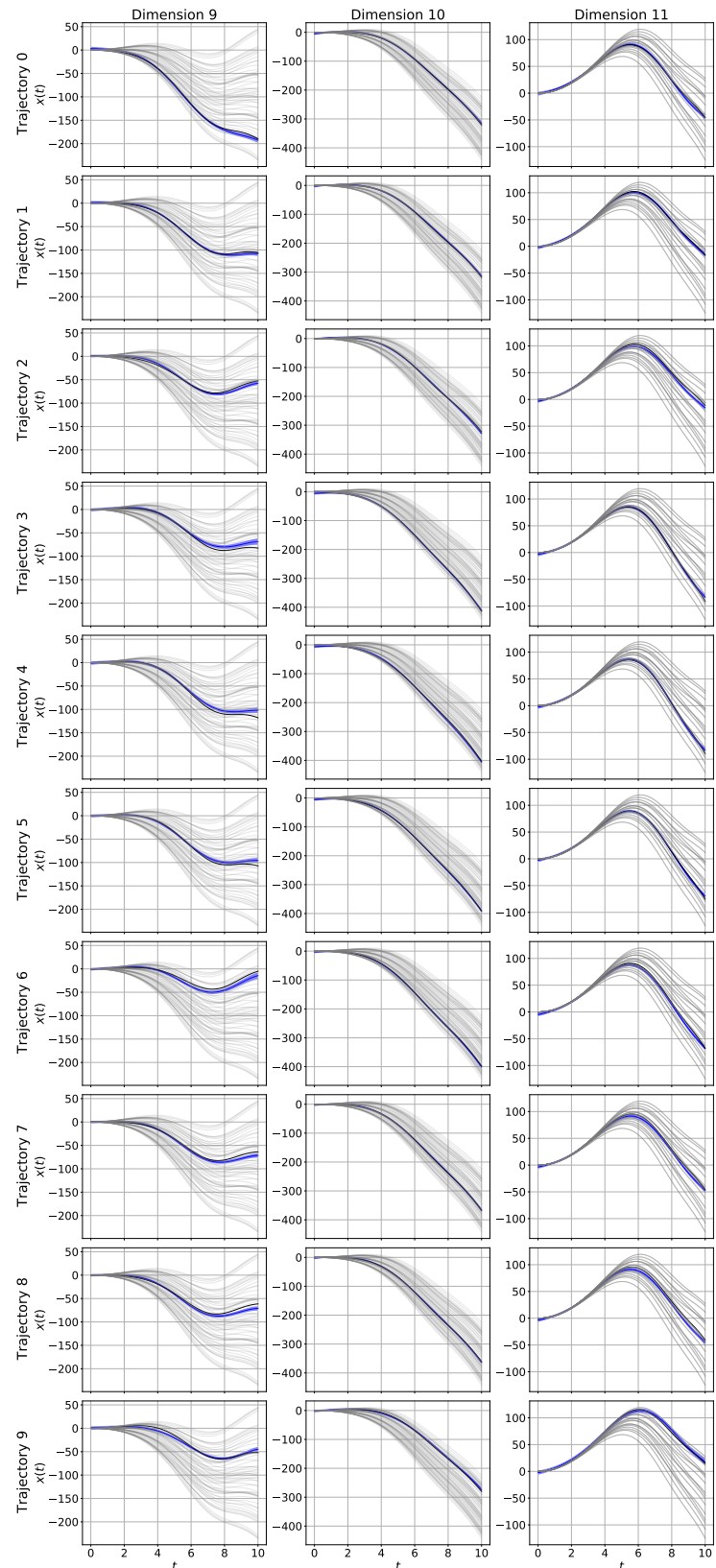

Figure 27: DGM's prediction on 10 randomly sampled test trajectories of QU 64, for state dimensions 9-11.

### E.3 Comparison with parameteric integration

In this subsection, we compare DGM against SGLD and SGHMC in the parametric setting, i.e. where we assume access to the parametric form of the true dynamics $f(x, \theta)$. Despite serious tuning efforts outlined in Appendix C.2, we were unable to make SGLD and SGHMC perform on any multitrajectory experiments except for Lotka Volterra 100. As can be seen in Table 5, the sampling based methods seem to perform quite well. However, it should be noted that we were unable to get a stable performance without using ground truth information, as outlined in Appendix C.2. Given this caveat and the results of the non-parametric case in the main paper, we conclude the following. If strong and accurate expert knowledge is available that can be used to fix strong priors on simple systems, the sampling-based approaches are certainly a good choice. For more complex systems or in the absence of any expert knowledge, DGM seems to have a clear edge.

Table 5: Log likelihood of the ground truth of 100 points on the training trajectories. SGHMC and SGLD were provided with strong, ground-truth-inspired priors and received an extensive hyperparameter sweep using the ground truth as metric. Nevertheless, DGM performs decently in comparison, without using neither priors nor ground truth.

|        | Log Likelihood | | |
|--------|----------------|------------------|------------------|
|        | DGM            | SGLD             | SGHMC            |
| LV 1   | $1.98 \pm 0.18$  | $\mathbf{3.07 \pm 0.685}$ | $3.06 \pm 0.517$ |
| LO 1   | $-0.52 \pm 0.09$ | $\mathbf{2.01 \pm 0.548}$ | F |
| DP 1   | $2.16 \pm 0.13$  | $\mathbf{3.43 \pm 0.396}$ | $2.96 \pm 0.795$ |
| QU 1   | $0.71 \pm 0.07$  | $\mathbf{2.42 \pm 0.322}$ | $1.38 \pm 0.00$ |
| LV 100 | $1.85 \pm 0.11$  | $\mathbf{4.28 \pm 0.184}$ | $4.26 \pm 0.178$ |