# OpenReview forum: "Distributional Gradient Matching for Learning Uncertain Neural Dynamics Models"
_NeurIPS.cc/2021/Conference — NeurIPS 2021 Poster_

### Official Review · Reviewer_3nDp · 2021-07-01

**Rating:** 5
**Confidence:** 3

**Summary:**

The paper proposes to skip the integration of continuous-time models and directly learn a model that predicts the posterior of the state conditioned on the initial state and the desired time. To learn such a model with 'good' uncertainty estimation the authors mix GP's, neural networks and the wasserstein metric (I did not understand how the authors combine these approaches together). Within the experiments the paper shows that the log-likelihood is better for the proposed model on uncontrolled toy systems, e.g. Lotka Volterra, chaotic Lorenz, double pendulum and quadcopter.


**Main Review:**

First of all, I am an expert on learning dynamics models but not an expert for Bayesian models. My main problem with the paper is that I cannot understand the proposed approach. The authors described the problem and their approach. However, once the authors mix all concepts of bayesian networks, GP's, Wasserstein distance and stir it all together. I personally do not have any understanding how all of this plays together. Therefore, the paper needs a major revision that restructures the writing such that the reader gets a good understanding of the approach. A good approach is to provide the intuition of the proposed approach and only afterwards deep dive into the details. However, the authors skip the intuitive approach and directly deep dive into the theory and lose the reader in the process. Besides the writing, the performed experiments are not really impressive and only try a few toy domains. Even for the toy domains the experimental evaluation does not provide much insights except my number is bigger than yours. It would be beneficial to incorporate more qualitative evaluations and explain to the reader what to look out for. Furthermore, the authors motivate their approach using control and RL but then only model uncontrolled dynamics and never scale the proposed approach to interesting control problems. How does the model work for high-dimensional systems with contacts? Therefore, it would be necessary to evaluate the proposed approach on more complex domains such as walker or humanoid.

As the writing and structure of the paper needs a major revision and the performed experiments do not evaluate whether this model actually improves learned models for control, I cannot recommend the paper for acceptance.

**Time Spent Reviewing:**

3

---

> ### Author Response · Authors · 2021-08-09
> **Response to Reviewer 3nDp**
>
> Thank you for your review and for providing us with an interesting perspective from the (non-Bayesian) dynamics learning community.
> Below we would like to restate our high-level ideas and address your concerns about the clarity and scale of the experiments.
>
> The key idea of our model is quite simple. To accommodate noisy observations, we smooth the observed data with a smoother. At the same time, we use the state and derivative estimates of the smoother to fit a probabilistic dynamics model. Here, the key point is that all of this happens on the level of distributions. This is important, since it allows us to learn uncertain dynamics, instead of obtaining point estimates. As assessed by reviewer u7cP, these are relevant questions for which the paper provides "good motivation". Similarly, we agree with reviewer zfk4, who states that our work "solves indeed a real bottleneck of existing approaches when used with NODE models (numerical integration)."
>
> We acknowledge that due to the difficulty of the task our method is rather complex.  We improved the clarity of the presentation based on your and the other reviewers' suggestions, which we greatly appreciate (see also the comment in the response to reviewer u7cP).
> In particular, following your advice, we more clearly emphasize the high-level idea in the introduction, such that the key points mentioned at the beginning of this comment become clearer.
>
> Given our focus on uncertainty quantification, in our experiments, we primarily compare with other approaches yielding uncertainty estimates, particularly based on approximate Bayesian inference.  Note that state-of-the-art models already struggle with "simpler" systems, without contact forces or other additional complexities, as demonstrated in our experiments.
> We are not aware of time-continuous alternatives that reliably quantify uncertainty about dynamics for systems like the walker or the humanoid.
> This is acknowledged, e.g., by reviewer oFcP, who states: "The numerical examples seem impressive, the quadrocopter system in particular being highly non-trivial."
>
> Since the state-of-the-art comparisons already struggle significantly with "simpler" systems, we strongly believe that our work provides a significant step forwards, as best summarized by reviewer zfk4, who believes "that the work present could be quite significant in the field, both for practitioners and for researchers who could build on these ideas." We would thus kindly ask you to reconsider your score.

---

> > ### Author Response · Authors · 2021-08-24
> > **Short follow-up**
> >
> > We would like to thank reviewer 3nDp again for their helpful comments and feedback. Please let us know if the above response was sufficient and in case there are no further questions, we would appreciate if you could update your score.

---

> > > ### Comment · Reviewer_3nDp · 2021-08-24
> > > **Short follow-up**
> > >
> > > I cannot increase my score as my main concern for this paper is the presentation of the material, which is not very intuitive or clear for somebody without deep knowledge of the Bayesian approaches. I am aware that you cannot adapt the paper during the rebuttal and hence, improve the clarity of the paper. I cannot update my score solely based on promises. However, I don't think that my vote will affect the final decision and I am not opposing an acceptance.
> > >
> > > And a minor side point. I can read the other reviews. Therefore, there is no need to copy the reviews of the other reviewers to my rebuttal reply. This just creates more useless repetitions. This new rebuttal style (which somehow got very popular within the recent conferences) has become way too annoying.

---

### Official Review · Reviewer_zfk4 · 2021-07-10

**Rating:** 7
**Confidence:** 3

**Summary:**

The paper tackles the problem of learning nonlinear dynamical systems from historical trajectories, while estimating predictive uncertainties. They propose a new method based on a Gaussian process smoother which is trained with maximum likelihood to fit the data of all training trajectories jointly, while constrained to follow in some sense the underlying dynamics. The latter is performed thanks to a regularizer forcing the smoother’s gradients distribution to match the  distribution of gradients of the dynamics model over a fixed set of support points. The Wasserstein-2 distance is approximated to evaluate the distribution discrepancies. Then, instead of deriving uncertainty estimates over the predicted parameters, as done in standard Bayesian methodology, uncertainty is directly estimated over the predicted states. For this, a Gaussian distribution is used to model  probability of the states derivatives conditioned on the states. Thanks to these tricks and modeling choices, the authors avoid to have to integrate the complex dynamics model, unlike previous Monte Carlo methods. The proposed approach is show-cased and compared to sampling-based methods in several experiments with simulated data, including overparametrized linear dynamics, as well as single and multi-trajectory cases of 4 well-known dynamical systems. An ablation study is also presented showing the benefits of joint smoothing and gradients distribution matching regularization on one of the four simulated datasets.

**Limitations And Societal Impact:**

I can't see any societal impact related to this publication.

Also, the authors don't seem to have addressed the limitations of the proposed approach.

**Main Review:**

Post-rebuttal additions
------------------------------

I would like to thank the authors for their detailed answers, which address all concerns I had.

-----------------------------------------------------------------------------------

Originality
--------------

The paper seems original to me and the related work seems quite well referenced to me.

Significance
----------------

The proposed approach is very complex, but not more than existing ones, so I believe it could be a good alternative to continuous-time nonlinear system identification requiring uncertainty estimates. It solves indeed a real bottleneck of existing approches when used with NODE models (numerical integration). Also, instead of delivering parameter uncertainties as previous methods, is focuses on direct estimation of state uncertainties, which is what matters at the end of the day. Hence I believe that the work present could be quite significant in the field, both for practitioners and for researchers who could build on these ideas.

Quality
----------

Overall quality of the explanations and experiments is quite good, as the method was show cased in varied settings and compared to relevant state-of-the-art alternatives. Also, the code seems clean (+1 for reproducibility). I do have some remarks though concerning some points that do not seem explained in the paper, as well as a few questions listed below:

1. While it is said in section 2 that trajectories can be sampled at non-equality spaced times, all simulated experiments use uniform time grids, which is a shame.
2. How are the supporting points chosen and how do they influence the estimates’ quality? For example, having a set of supporting points not well spread or too small should lead to degraded performance should it not?
3. In equation 12 and line 196, it is said that dynamics noise $epsilon$ is drawn one per trajectory, while in the problem statement, observations $y$ are supposed to have i.i.d additive noise (one per sample in each trajectory). While I realize these quantities are not the same (one is in the states and the other in their derivatives), I wonder why make these different modeling choices and not also have a different disturbance realization for each instant in the dynamics model.
4. Concerning the ablation study, you say on line 330 that by comparing the rows of Figure 1 it can be seen that joint smoothing of multiple trajectories helps. While I do see it clearly when comparing rows of the second column (i.e. with the dynamics regularizer), I see the opposite when looking at the first column. Namely, joint smoothing seems to lead a noisier and less accurate predicted trajectory, with a larger confidence region. As this is very curious, since it contradict the text, I think it should be commented and also that the sentence should be clarified (by saying that improvement is visible on the second column for example).
5. How many training and testing trajectories are used to compute results for figure 2?
6. Likewise, while it is said that 5 to 10 points per trajectory are used to train the models in the multi-trajectory experiment, how many trajectories are used? Is that the meaning of the numbers next to the systems considered in the first column of Table 2? If that’s the case, please add it to the legend or paragraph text.
7. Could you please comment the fact that SGLD and SGHMC methods outperform DGM for configurations 3.3 and 3.6.3 on figure 2?
8. Could you please develop on the reason SGLD failed on the quadcopter single trajectory experiment, leading to NaNs? Did you try to understand what these NaNs meant?

Clarity
---------

1. Although the paper is well written, the proposed method is very complicated and difficult to grasp (I had to read the paper many times before I started to get it). I cannot say it is not well explained though, but it might help to put more structure in section 3 (replace paragraphs by subsections + parapgraphs for example). Also, adding a diagram summarizing how the many elements of the method interact with each other could help a lot I believe.
2. Furthermore, I found some explanation was missing concerning the certainty-equivalence approximation of the Wasserstein distance. Maybe add that the expectation you are approximating is the one in (15) and that this approximation is carried in the second line go (16).
3. In the paragraph Final loss (page 6), the first data fitting term of the loss is denoted quite differently than in equation (8). It took me some time to realize they were the same, so I think some referencing and more consistent notation could help with the clarity here.
4. In line 285, you denote the model used to fit the dynamics with the same notation as in line 282, where you explain the real dynamics that generated the data. I find this confusing and believe that less ambiguous notation here would help with the clarity (by using for example the dynamics function $f$ introduced in (1)).
5. I could not find any introduction of the abbreviations SGLD and SGHMC.
6. On line 164, I believe $z_i:=$ should not be there, as it is defined differently in the text and in equations below line 166.
7. Sentence at lines 119-120 has a problem as the word maps is repeated twice.
8. Line 244: another -> each other

**Time Spent Reviewing:**

8

---

> ### Author Response · Authors · 2021-08-09
> **Response to Reviewer zfk4**
>
> **\(C\)** *How are the supporting points chosen and how do they influence the estimates’ quality?* **\(R\)** As mentioned in the Appendix, Section B.4, we choose 30 equidistant support points on every trajectory in the multi-trajectory case and in the single-trajectory case we choose the same number of supporting points as we have the observation points. This is set to be a default, found by conducting an ablation study where we selected 5, 10, 30, 50, and 100 supporting points in the multi-trajectory case. We observed that with too few supporting points we don't get good predictions, while for 30 and more we obtain quite similar performance.
>
> **\(C\)** *Equation 12 and line 196.* **\(R\)** For the derivative model, we cannot treat the noise independently. If we were to sample $\mathbf{\epsilon}$ at every instant of the dynamics model, i.e. at every continuous-time point, we would in principle recover an SDE. SDEs are not differentiable almost everywhere, invalidating the fundamental premise of gradient matching. Instead, $\mathbf{\epsilon}$ is treated as a weight vector parametrizing an ensemble of deterministic neural ODEs. The uncertainty is then captured by this ensemble, not by the random variable itself.
>
> **\(C\)** *Ablation study.* **\(R\)** We propose to replace the second and third sentence of that paragraph with "While learning a joint smoother enables the model to share data between trajectories, this data is not used effectively without a dynamics model. Only after introducing the additional regularization imposed by a dynamics model, the smoother is able to leverage this additional knowledge to reduce the uncertainties even between observation points." Does this incorporate your concerns sufficiently?
>
> **\(C\)** *How many training and testing trajectories are used to compute results for figure 2?* **\(R\)** As mentioned in the Appendix, Section C.1, we train and test on one trajectory, but repeat that experiment 10 times. For clarity, we will add a reference to C.1 in line 296.
>
> **\(C\)** *Number of trajectories per system.* **\(R\)** As you correctly inferred, the number of trajectories is represented by the number next to the systems considered in Table 2. In the caption of Table 2, we will add as the last sentence "The number following the system name denotes the number of trajectories in the training set.".
>
> **\(C\)** *SGLD and SGHMC methods outperform DGM.* **\(R\)** For simple, linear systems, numerical integration performs well. In this case, the number of parameters is small and the posterior distribution over parameters is unimodal. This favors the MCMC approaches (SGHMC and SGLD), which will converge quickly. For more complicated systems, parameter spaces become larger, the posterior distributions more complicated and small parameter deviations might already lead to dynamics that cannot be numerically integrated, see also the next comment.
>
> **\(C\)** *SGLD failed on Quadrocopter.* **\(R\)** The quadrocopter dynamics are quite sensitive with respect to their parameters. The NaNs are created if we integrate slightly wrong dynamics, as then the norm of the state blows up. While this is a problem for all approaches, SGLD seems to have been especially affected in our case.
>
> **\(C\)** *Structure in Section 3.* **\(R\)** As detailed in our comment to reviewer u7cP, we changed the presentation of deep Gaussian processes in Section 3. Do these changes address your concerns sufficiently?
>
> **\(C\)** *Explanation concerning the certainty-equivalence approximation.* **\(R\)** We will add the statement that we perform certainty-equivalence approximation in the second line of the Equation (16) in the amended version of the paper.
>
> **\(C\)** *Different notation for $\mathcal{L}\_{\text{data}}$.* **\(R\)** Agreed, we completely dropped $p(\mathbf{y}| \mathbf{\varphi})$ and now exclusively use $\mathcal{L}\_{\text{data}}$. Thanks for raising this confusion.
>
> **\(C\)** *Confusing definition of a model in line 285.* **\(R\)** We will update the notation for the ansatz in the line 285 to $f(\mathbf x, \mathbf \theta) = \mathbf B\mathbf x$ as proposed.
>
> **\(C\)** *Full names for SGLD and SGHMC.* **\(R\)** We will add the full names and references to the methods where they are first mentioned in the amended version of the paper.

---

> > ### Comment · Reviewer_zfk4 · 2021-08-18
> > **Further comments**
> >
> > Thanks for your answer. My concerns were all well addressed, but the one concerning the structure of section 3. I don't think that changing the notation of the GP mean and covariance matrix solves the problem with the structure of the section. As mentioned by other reviewers, it is very difficult to understand the proposed approach. I still think that subsections in different levels would help to understand how the many elements presented are grouped and I'm also still in favor of a diagram depicting how they interact.
> >
> > The higher level description that you propose to add to the introduction in your answer to reviewer 3nDp could help too. Would it be possible to paste here the changes you have in mind?
> >
> > Also, I believe there was a mistake in your answer because I could not find any reference to experiments repeated 10 times in section C.1.

---

> > > ### Author Response · Authors · 2021-08-20
> > > **Response to Further comments**
> > >
> > > Thank you for your response, we will gladly outline the changes we have in mind.
> > >
> > > In the introduction, we plan to add a paragraph giving a high-level overview of the algorithm, as drafted here (https://ibb.co/wW8FdRn). The structure in the diagram resembles the paragraphs in Section 3, where the dense presentation in the form of paragraphs was necessary due to space constraints. However, given that the final version will have less space restrictions, we agree that it it is a good idea to turn these paragraphs into subsections and already reference them in the short overview paragraph written in the introduction.
> > >
> > > As for the reference in C.1., we would add at line 296, after the sentence "In Figure 2, we show the mean and standard deviation of the log likelihood of the ground truth over 10 different noise realizations" the new sentence
> > > "The exact procedure for one noise realization is described in the appendix, Section C.1."
> > > The ten noise realizations are already mentioned in the paper, so that the reader can understand where a standard deviation is coming from. Everything else is pushed to the appendix.

---

### Official Review · Reviewer_oFcP · 2021-07-12

**Rating:** 7
**Confidence:** 2

**Summary:**

This paper develops a method called distributional gradient matching for learning unknown differential equations from data. The method is based on a combination of a neural ODE model and a Gaussian process model (with a deep covariance kernel) which is encouraged to produce solutions which follow the neural ODE model by introducing a regularisation term based on Wasserstein loss. Numerical examples show that the proposed method outperforms some existing alternatives.

**Ethical Concerns:**

-

**Limitations And Societal Impact:**

Limitations caused by the cubic computational cost of GP regression are mentioned.

**Main Review:**

I am not an expert on learning ODEs, but the literature reviews in Sections 1 and 5 do make it seem like the approach is new. It would be interesting to have more commentary on how the use of the GP model in this paper differs from prior work - I presume that most prior work does not use a deep GP model.

The numerical examples seem impressive, the quadrocopter system in particular being highly non-trivial. In these examples the proposed method clearly outperforms some existing methods, both in terms of accuracy and speed. I lack expertise to say if the two MC-based algorithms, SGLD and SGHMC, that the comparisons are made to are representative of the state-of-the-art in this setting.

In my opinion the main weakness of this paper is that the presentation of the method is often quite confusing and the notation not always properly explained:
- Line 141: What does \dot{\mathcal{X}} stand for?
- p4: It is not told what the GP model is supposed to be modelling, just that "we define a Gaussian process".
- p4: The use of a mapping \phi is not standard in GP regression (at least yet), so it would be good to make it clearer that this part of the GP model comes from Wilson et al. (2016).
- p4: To call \phi a "feature map" seems potentially confusing if one is used to defining covariance kernels via inner products of feature maps.
- p4: Is the GP covariance kernel supposed to be potentially different for each k (and hence the notation \mathcal{K}_k)? If so, is it intentional that there is only a single prior mean function?
- Remarks on p6 that it has "trainable parameters" and in Appendix B.2 that it is "deep" appear to be the extent of exposition on the GP prior mean function \mu. More should perhaps be said.
- What are t and t_{supp} in the definitions on p4?
- \mathcal{L}_{data} defined in Equation (8) is never used afterwards; it is denited p_{GP}(y | \varphi) later.
- Is \mu_{S,supp} on p6 equal to \mu_S in Equation (7)?
- It would be helpful to tell explicitly that the Wasserstein loss in Equation (16) will be used as \mathcal{L}_{dynamics}.

I am not sure if the paper contains sufficient details to replicate the experiments.

Other minor comments:

- The sentence on lines 119-121 is quite awkward. There is also an extra closing parenthesis on line 120.
- Line 131: "smother-dependent"
- I believe the dot over \mu in Equation (7) is larger than that in the equation between lines 166 and 167.
- The GP prior mean needs to be subtracted from second term in Equation (10).
- Transpose notation is not consistent: E.g., Equations (10) and (11) use "T" while (6) and (7) use "\top".
- \mathcal{D} in Equation (9) should probably have subscript k.
- That the noise variance matrix is \Sigma_\epsilon given that Equation (12) has a different random variable \epsilon.
- Line 196: \epsilon should probably be bold.
- Lines 216-17: "regularizaton"
- Line 223: "indepdent"
- "Lahdesmaki" - "Lähdesmäki" in one of the references


**Time Spent Reviewing:**

6

---

> ### Author Response · Authors · 2021-08-09
> **Response to Reviewer oFcP**
>
> **\(C\)** *It would be interesting to have more commentary on how the use of the GP model in this paper differs from prior work.* **\(R\)** The key novelty of our work does not lie in the way that the GP model is set up, but in the inference step. Thus far, there were three "philosophies". Most works that investigate Bayesian inference, e.g., Calderhead et al. (2009), Dondelinger et al. (2013), Wenk et al. (2019), use MCMC sampling schemes, which are already quite slow for <10 parameters. In the context of Bayesian inference, there is also one work using variational inference, i.e. Gorbach et al. (2017), but it is restricted to a very specific, local linear dynamics structure. Finally, in the context of frequentist inference, Wenk et al. (2020) infers point estimates by solving an optimization problem, which is fast but does not provide uncertainty estimates. Our method provides uncertainty estimates by solving an optimization problem, which combines the best of both worlds. In this sense, the crucial component is not the GP model itself, but the way inference is done. Since none of the previously mentioned GP-based approaches is directly suitable for neural ODEs, we decided to focus the discussion more on approaches that can already do inference in the context of neural ODEs. Do you feel like the previous discussion would add value to the paper?
>
> **\(C\)** *State of the art MC approaches.* **\(R\)** Regarding MC approaches we followed a recent study of Dandekar et al. (2021). They identify NUTS, SGHMC and SGLD as the best candidates for Bayesian inference, with NUTS trailing behind the other two. In our experiments, we were able to confirm this observation and thus chose SGHMC and SGLD as the state-of-the-art representatives of the MC community.
>
> **\(C\)** *What does $\dot{\mathcal{X}}$ stand for?* **\(R\)** $\dot{\mathcal{X}}$ is the same as $\dot{\mathcal{X}}\_{\text{supp}}$. We changed the notation while writting the paper and instead of $\dot{\mathcal{X}}\_{\text{supp}}$ should be $\dot{\mathcal{X}}$ already in the definition. We use $\dot{\mathcal{X}}$ everywhere else. We will amend this in the updated version of the paper, thanks for spotting that.
>
> **\(C\)**  *Presentation of Deep Gaussian processes and GP mean and covariance kernel different for each $k$.* **\(R\)** We changed the presentation of Deep Gaussian processes, see comment to the reviewer u7cP. Does this address your concerns sufficiently? The presentation was now chosen to reflect that everything is independent across dimensions, while the implementation details (including shared components) have been pushed completely to the Appendix, Section B.2.
>
> **\(C\)** *What are $\mathbf t$ and $\\mathbf t_{\text{supp}}$ in the definitions on page 4?* **\(R\)** $\mathbf t$ stands for concatenated observation times over every time and every trajectory and $\mathbf t_{\text{supp}}$ stands for concatenated times at which we match the gradients of the distributions. For clarity, we added the last statement in the updated version of the presentation of Deep GPs, as referenced in the previous comment.
>
> **\(C\)** *$\mathcal{L}_\text{data}$ definition.* **\(R\)** Thanks, we change $p(\mathbf{y}|\mathbf{\varphi})$ to $\mathcal{L}\_{\text{data}}$ where the final loss is introduced and dropped $p(\mathbf{y}|\mathbf{\varphi})$ completely for simplicity.
>
> **\(C\)** *Is $\mu_{S,supp}$ on page 6 equal to $\mu_S$ in Equation (7)?* **\(R\)** We changed $\mathbf{\mu}\_S$ to $\dot{\mathbf{\mu}}\_{S, \text{supp}}$ to reflect that these are two completely different quantities. $\dot{\mathbf{\mu}}\_{S, \text{supp}}$ denotes the mean over derivatives at the supporting points, while $\mu\_{S,supp}$ denotes the mean of the states at the supporting points.
>
> **\(C\)** *$\mathcal{D}$ in Equation (9) should probably have subscript $k$.* **\(R\)** Since the dimensions are treated independently, this does not make a difference. However, we see this could help with readability and added it accordingly.
>
> **\(C\)** *That the noise variance matrix is $\Sigma_\epsilon$ given that Equation (12) has a different random variable $\epsilon.$* **\(R\)** Thanks, we changed $\Sigma_{\mathbf \epsilon}$ to $\Sigma_{\text{obs}}$.
>
> **References**
>
> Calderhead, B., Girolami, M., and Lawrence, N. D. (2009). Accelerating bayesian inference over nonlinear differential
> equations with gaussian processes. In Advances in neural information processing systems, pages 217–224. Citeseer.
>
> Dandekar, R., Chung, K., Dixit, V., Tarek, M., Garcia-Valadez, A., Vemula, K. V., and Rackauckas, C. (2021). Bayesian neural ordinary differential equations. Symposium on Principles of Programming Languages, POPL.
>
> Dondelinger, F., Husmeier, D., Rogers, S., and Filippone, M. (2013). Ode parameter inference using adaptive gradient matching with gaussian processes. In Artificial intelligence and statistics, pages 216–228. PMLR.
>
> Gorbach, N. S., Bauer, S., and Buhmann, J. M. (2017). Scalable variational inference for dynamical systems. In Guyon, I., Luxburg, U. V., Bengio, S., Wallach, H., Fergus, R., Vishwanathan, S., and Garnett, R., editors, Advances in Neural Information Processing Systems, volume 30. Curran Associates, Inc.
>
> Wenk, P., Abbati, G., Osborne, M. A., Schölkopf, B., Krause, A., and Bauer, S. (2020). Odin: Ode-informed regression for parameter and state inference in time-continuous dynamical systems. In Proceedings of the AAAI Conference on Artificial Intelligence, volume 34, pages 6364–6371.
>
> Wenk, P., Gotovos, A., Bauer, S., Gorbach, N. S., Krause, A., and Buhmann, J. M. (2019). Fast gaussian process based gradient matching for parameter identification in systems of nonlinear odes. In The 22nd International Conference on Artificial Intelligence and Statistics, pages 1351–1360. PMLR.

---

### Official Review · Reviewer_u7cP · 2021-07-16

**Rating:** 7
**Confidence:** 4

**Summary:**

This paper presents a novel approach to model and capture the uncertainty of dynamical systems by learning from trajectory data. The method is composed of a Gaussian process smoother model, used for predictions, and a neural dynamics model, used for training. The novelty in the method comes from the use of the distributions over time-derivatives of the state vectors (via a Wasserstein-distance penalty between the distributions from the smoother and the dynamics model) during the training process. Basing the approach on integrals over the state space, instead of over the model parameters space, allows for computational efficiency and more robustness when compared to previous sampling-based approaches, as evidenced by experiments.

**Limitations And Societal Impact:**

Some of the scalability limitations are explored in the appendix, and no negative societal impact is foreseen by the authors.

**Main Review:**

The paper is mostly well written, providing a good motivation, an appropriate use of citation and interesting insights in the derivation of the method. However, there are a few issues, especially regarding methodological details, which make it hard to understand the method and to assess its soundness.

Major issues:
- The description of the methodology is somewhat confusing in a few important aspects.
For instance, how does the set of "supporting gradients" play a role in the estimation of these two distributions? Are they random variables to be inferred as part of $\dot{\mathcal{X}}$? Or are they additional observations which will be used somehow in the calculation of the losses?
- Regarding $\mathbf{z}$ in Eq. 6, I get the idea, but it's a bit confusing at first-reading to use both $\mathbf{z}$ and the $(\mathbf{x}, t)$-tuples as inputs to the GP when the feature map "phi" is supposed to be part of the kernel. It'd be better to consider $k := k \circ \phi$ (composition), since the composition of a kernel with an input map is still a kernel, and then represent the inputs as simply $(\mathbf{x},t)$, without the extra $\mathbf{z}$ notation.
- In line 196, it is mentioned that the (instant) state noise vector in Eq. 12 is drawn only once per rollout. Does it mean that the noise term epsilon is drawn once and repeated for all time points along a trajectory? If so, the state noise is not independent, but actually correlated across time and that would possibly not allow the model in Eq. 12 to properly capture uncertainty in the state transitions.
- I'm not sure about the effects of this approximation in (14), since later on the difference between $p_D$ and $p_S$ will be minimised. Having $p_D$ as a function of $p_S$ might have negative side effects. Any ideas on the drawbacks?
- The certainty-equivalence approximation (see line 220) discards the uncertainty captured by the GP model in the expectation in Eq. 15. Why not using a quadrature approximation to compute Eq. 15, like the unscented transform, or anything else which could capture the states covariance matrix on the inputs?
- In the experiments with SGLD and SGHMC, how is the likelihood for these sampling-based methods computed? These methods don't directly produce a probability density, only empirical approximations.
- The experimental evaluation could be more complete if they included comparisons against methods which encode (inexact) prior knowledge about the dynamics, such as simulator-based inference approaches [e.g., A, B, C, below]. The relationship to these methods is also not discussed in the related work section. In my view, the proposed approach could be combined with simulator-based models by using a physics-informed mean function for the GP smoother, which is not explored/discussed in the paper.

Minor issues:
- Parametric vs. Non-parametric: Neural networks are "parametric" models, though in the main paper and in the appendix they are referred to as "non-parametric" models. The only non-parametric model in this work is the GP.
- The equation for $\mathbf{z}_i$ in line 164 is missing $\phi$.
- I couldn't find details of this weight sharing scheme in Sec. 4, despite the mention in line 200.
- Preliminary experiments: Are these results available? If not, it'd be better to add them to the appendix or give more details/references to back it up. For example, the Wasserstein GANs paper provides some experiments and theoretical justification on why the Wasserstein distance is more appropriate than the KL divergence for models over high-dimensional space, which usually have their data distribution concentrated on a (latent) lower-dimensional subspace.
- Experiments: At the first mention of SGLD, SGHMC and NUTS, please spell out and/or add references to these methods for readers who are unfamiliar with the literature. Also, please, consider adding a reference to NDP in line 265.
- Experiments: Line 306, any idea why SGLD failed with the Quadrocopter 1?
- Please, add (foot)note on what is an "Ansatz". At first, I thought it was a typo, as I was unfamiliar with the term.

References:

[A] Cranmer, Kyle, Johann Brehmer, and Gilles Louppe. 2020. “The Frontier of Simulation-Based Inference.” Proceedings of the National Academy of Sciences.

[B] Ramos, Fabio, Rafael Carvalhaes Possas, and Dieter Fox. 2019. “BayesSim : Adaptive Domain Randomization via Probabilistic Inference for Robotics Simulators.” In Robotics: Science and Systems (RSS). Freiburg im Breisgau, Germany.

[C] Okada, Masashi, and Tadahiro Taniguchi. 2019. “Variational Inference MPC for Bayesian Model-Based Reinforcement Learning.” In 3rd Conference on Robot Learning (CoRL 2019).

**Time Spent Reviewing:**

4

---

> ### Author Response · Authors · 2021-08-09
> **Response to Reviewer u7cP**
>
> **\(C\)** *How does the set of "supporting gradients" play a role in the estimation of these two distributions?* **\(R\)** Based on the smoother and the dynamics model we compute the distribution of the *supporting gradients* at the *supporting locations*. They are random variables and are not additional observations. We don't observe supporting gradients but compute it's distribution on two different ways and then minimize Wasserstein 2-distance between the computed distributions.
>
> **\(C\)** *Confusing $\mathbf z$ in Eq. 6 and in the paragraph above.* **\(R\)** We changed the presentation of the deep Gaussian processes in the following way (line 161 to line 167). Does this address your concerns sufficiently?:
>
> ---------------------
> We define a Gaussian process with a differentiable mean function $\mu(\mathbf{x}\_m(0), t\_{n,m})$ as well as a differentiable and positive-definite kernel function $\mathcal{K}\_{\text{RBF}}(\mathbf{\phi}(\mathbf{x}\_m(0), t\_{n, m}), \mathbf{\phi}(\mathbf{x}\_{m'}(0), t\_{n', m'}))$. Here, the kernel is given by the composition of a standard ARD-RBF kernel (Rasmussen, 2004) and a differentiable feature extractor $\mathbf{\phi}$ parametrized by a deep neural network, as introduced by Wilson et al. (2016).
> Following Solak et al. (2003), given fixed $\mathbf{x}\_{\text{supp}}$,
> we can now calculate the joint density of $(\dot{\mathbf{x}}^{(k)}\_{\text{supp}}, \mathbf{y}^{(k)})$ for each state dimension $k$. Concatenating vectors accordingly across time and trajectories, let
> $$
>  \qquad  \qquad  \qquad  \qquad \mathbf{\mu}^{(k)} := \mu^{(k)}\left(\mathbf{x}(0), \mathbf{t}\right),  \quad \\,\\,\\,\\,
> \dot{\mathbf{\mu}}^{(k)} := \frac{\partial}{\partial t} \mu^{(k)}\left(\mathbf{x}\_{\text{supp}}(0), \mathbf{t}\_{\text{supp}}\right),
> $$
> $$
> \qquad  \qquad  \qquad  \qquad\mathbf{z}^{(k)} := \phi^{(k)}(\mathbf{x}(0), \mathbf{t}), \quad \\,\\,\\,\\,
> \mathbf{z}^{(k)}\_{\text{supp}} := \phi^{(k)}(\mathbf{x}\_{\text{supp}}(0), \mathbf{t}\_{\text{supp}}),
> $$
> $$
> \qquad  \mathbf{\mathcal{K}}^{(k)} := \mathcal{K}\_{\text{RBF}}^{(k)}(\mathbf{z}^{(k)}, \mathbf{z}^{(k)}), \quad
> \dot{\mathbf{\mathcal{K}}}^{(k)} :=\frac{\partial}{\partial t\_1}\mathcal{K}\_{\text{RBF}}^{(k)}(\mathbf{z}^{(k)}\_{\text{supp}}, \mathbf{z}^{(k)}), \quad
> \ddot{\mathbf{\mathcal{K}}}^{(k)} := \frac{\partial^2}{\partial t\_1 \partial t\_2} \mathcal{K}\_{\text{RBF}}^{(k)}(\mathbf{z}^{(k)}\_{\text{supp}},
> \mathbf{z}^{(k)}\_{\text{supp}}).
> $$
> ---------------------
>
> **\(C\)** *In line 196, it is mentioned that the (instant) state noise vector in Eq. 12 is drawn only once per rollout.* **\(R\)** Intuitively, $\mathbf{\epsilon}$ should not be seen an independent, instant state noise vector in the sense of an SDE. SDEs are not differentiable almost everywhere, invalidating the fundamental premise of gradient matching. Instead, $\mathbf{\epsilon}$ should be seen as a weight vector parametrizing an ensemble of deterministic neural ODEs, leading to an ensemble of deterministic trajectories. This allows us to capture the uncertainty directly in the state space via this ensemble, not by the random variable itself, greatly improving tractability w.r.t. state of the art methods.
>
> **\(C\)** *Approximation in (14).* **\(R\)** This approximation forms the core of many gradient matching schemes, including ODIN (Wenk et al., 2020), RKG3 (Niu et al., 2016), AGM (Dondelinger et al., 2013), FGPGM (Wenk et al., 2019), Calderhead (Calderhead et al., 2009) and many more. It is crucial in avoiding the numerical integration bottleneck, where the smoother plays the role of the integrator. Implicitly, we assume that the states obtained via (integrating) the dynamics model are sufficiently close to the states obtained by the smoother. This is reasonable, since we enforce via (15) that the derivatives of the dynamics and smoother match, and thus the integrated derivatives should be close as well. This approximation can be inaccurate in two cases. First, if the smoother model becomes degenerate, i.e. if it fails to capture the data, the state estimates of the smoother should not be trusted. Second, if the smoother does not capture the correct shape of the trajectory in between the supporting points, e.g. if not enough supporting points are chosen, the mapping between states and derivatives of the smoother becomes inaccurate. Thus, even for perfectly matched derivatives, the state estimates might be off. In both cases, (15) will add some regularization, but the model will ultimately fail. However, we never observed this in our experiments and we doubt that for such cases, any integration-free approach could work.
>
> **\(C\)** *The certainty-equivalence approximation (see line 220).*  **\(R\)** We agree that more involved approximations could be an interesting topic for future work and the unscented transform is an interesting candidate. Since the certainty-equivalence approximation already works quite well, we opted for simplicity.
>
> **\(C\)** *Likelihood for SGLD and SGHMC.* **\(R\)** We produce several trajectories by first sampling parameters of the ODE and then integrating the resulted dynamics (see also Appendix, C.3). The log-likelihood is then obtained by approximating the samples with a Gaussian distribution.
>
> **\(C\)** *Comparisons against methods which encode (inexact) prior knowledge about the dynamics.* **\(R\)** Prior physical knowledge could be introduced both on the level of the GP mean (which probably requires integration or simulation) as well as on the level of the dynamics model (more in line with the integration free spirit of our work). The case of white box dynamics is showcased in the Appendix, Section E.3, where we show an increase in log-likelihood for all methods if the true parametric form is known. However, to avoid a subjective discussion about how much prior knowledge is admissible, we focus our work purely on black-box models. Nevertheless, we agree that an extension to gray box models is interesting and relevant and should be investigated in future work, especially in the context of applications. We will mention this and the approaches you mentioned in our conclusion section.
>
> **\(C\)** *Details of this weight sharing scheme.* **\(R\)** We incorrectly pointed to Section 4. We describe the weight sharing scheme in Appendix B.1 and B.2. We will update the pointer in the updated version of the paper.
>
> **\(C\)** *Comparison between KL divergence and Wasserstein 2-distance.* **\(R\)**The main issue we experienced with the KL divergence was numerical instability. We found this to be due to the inversion of the covariance matrices, which often became ill-conditioned during the optimization process. Such inversion is not needed for calculating the Wasserstein distance. We agree that the statement in line 214 is a bit vague and will make it more concrete, reading
> "[..] closed-form representation. Furthermore, since no covariance matrices need to be inverted, it proved to be numerically superior to similar measures like [...].
>
> **\(C\)** *Full name and reference for SGLD, SGHMC and NUTS.* **\(R\)** Thanks for noticing, we will add a full name and reference to the method in the amended version of the paper.
>
> **\(C\)** *SGLD fails for Quadrocopter 1.* **\(R\)** The quadrocopter dynamics are quite sensitive with respect to their parameters. If we integrate slightly wrong dynamics it happens that the norm of the state blows up and then we obtain NaNs. While this is a problem for all approaches, SGLD seems to have been especially affected in our case.
>
> **References**
>
> Calderhead, B., Girolami, M., and Lawrence, N. D. (2009). Accelerating bayesian inference over nonlinear differential equations with gaussian processes. In Advances in neural information processing systems, pages 217–224. Citeseer.
>
> Dondelinger, F., Husmeier, D., Rogers, S., and Filippone, M. (2013). Ode parameter inference using adaptive gradient matching with gaussian processes. In Artificial intelligence and statistics, pages 216–228. PMLR.
>
> Niu, M., Rogers, S., Filippone, M., and Husmeier, D. (2016). Fast parameter inference in nonlinear dynamical systems using iterative gradient matching. In International Conference on Machine Learning, pages 1699–1707. PMLR.
>
> Rasmussen, C. E. (2004). Gaussian Processes in Machine Learning, pages 63–71. Springer Berlin Heidelberg, Berlin, Heidelberg.
>
> Solak, E., Murray-Smith, R., Leithead, W. E., Leith, D. J., and Rasmussen, C. E. (2003). Derivative observations in
> gaussian process models of dynamic systems.
>
> Wenk, P., Abbati, G., Osborne, M. A., Schölkopf, B., Krause, A., and Bauer, S. (2020). Odin: Ode-informed regression for parameter and state inference in time-continuous dynamical systems. In Proceedings of the AAAI Conference on Artificial Intelligence, volume 34, pages 6364–6371.
>
> Wenk, P., Gotovos, A., Bauer, S., Gorbach, N. S., Krause, A., and Buhmann, J. M. (2019). Fast gaussian process based
> gradient matching for parameter identification in systems of nonlinear odes. In The 22nd International Conference on Artificial Intelligence and Statistics, pages 1351–1360. PMLR.
>
> Wilson, A. G., Hu, Z., Salakhutdinov, R., and Xing, E. P. (2016). Deep kernel learning. In Artificial intelligence and statistics, pages 370–378. PMLR.

---

> > ### Comment · Reviewer_u7cP · 2021-09-13
> > **Reply to authors**
> >
> > I would like to thank the authors for their effort in addressing my concerns, which were mostly addressed, so I'm raising my score.

---

### Author Response · Authors · 2021-08-09
**General Response**

Many thanks for your detailed comments and suggestions. We are happy to see that overall, our work is quite well received and most reviewers agree on the relevance and the quality of our work. To adequately address individual concerns and questions, we decided to respond to each review individually. For each concern we identified, we created a short summary (following a **\(C\)**) and then addressed it directly afterwards (following a **\(R\)**). Please let us know if you have any further questions or comments, we strongly appreciate your feedback.

---

### Decision · Program_Chairs · 2021-09-27

**Decision:**

Accept (Poster)

**Comment:**

Even if there was some spread in the review scores which called for discussion among the reviewers, in the end, the reviewer consensus was in favour of accepting this paper. Please, make sure to address their concerns in the reviews in the camera-ready. The concerns related to clarity, presentation, and notation (which you also comment on in your response) are of particular importance as these help ensure impact of your work.